# A Black Swan Hypothesis: The Role of Human Irrationality in AI Safety

**Hyunin Lee**[1]* **Chanwoo Park**[2] **David Abel**[3] **Ming Jin**[4]
[1]UC Berkeley, [2]MIT, [3]Google DeepMind, [4]Virgina Tech

## Abstract

*Black swan* events are statistically rare occurrences that carry extremely high risks. A standard view of black swans assumes that they originate from an unpredictable and changing environment; however, the community lacks a comprehensive definition of black swan events. To this end, this paper challenges that the standard view is *incomplete* and claims that high-risk, statistically rare events can also occur in unchanging environments due to human misperception of events' values and likelihoods, which we refer to as S-BLACK SWAN . We first carefully categorize black swan events, focusing on S-BLACK SWAN , and mathematically formalize the definition of black swan events. We hope these definitions can pave the way for the development of algorithms to prevent such events by rationally correcting limitations in perception.

## 1 Introduction

To successfully deploy machine learning (ML) systems in open-ended environments, these systems must exhibit robustness against *rare and high-risk events*, often referred to as *black swans* (Taleb, 2010). Achieving this robustness requires a deep and precise understanding of the origins of such events, which has been increasingly recognized as a critical factor for enabling ML algorithms to attain full control and make optimal decisions (Chollet, 2019; Silva & Najafirad, 2020; He et al., 2021; Li et al., 2023; Yang et al., 2024). Nevertheless, many contemporary ML systems remain vulnerable to black swans in real-world scenarios, as evidenced by automated trading systems that overreact to market anomalies (Kirilenko et al., 2017; Phillips, 2021; Stafford, 2022), unexpected bankruptcies (Wiggins et al., 2014; Akhtaruzzaman et al., 2023), the Covid pandemic (Antipova, 2020), and autonomous vehicles encountering unforeseen road or weather conditions (Tesla, 2021; Witman et al., 2023; Nordhoff et al., 2023).

In this paper, we argue that ML systems remain susceptible to black swan events, regardless of an algorithm's representation capacity or scalability, due to an AI community's *incomplete* understanding of the origins of these events. The prevailing belief in most algorithmic approaches to preventing black swan events (Prestwich, 2019; Artemenko et al., 2020; Devarajan et al., 2021; Wabartha et al., 2021; Bhanja & Das, 2024; Jin, 2024) is that such events primarily arise from *dynamic, time-varying* environments. We contend, however, that black swans can also emerge from *static, stationary* environments. To this end, we propose a new hypothesis on their origins:

> **Hypothesis 1.** Black swans can originate from misperceptions of an event's reward and likelihood, even within static environments.

---

*Corresponding author: hyunin@berkeley.edu

To warmly introduce our new hypothesis, consider the bankruptcy of Lehman Brothers, widely recognized as the most significant black swan event in the financial industry (Wiggins et al., 2014). A strong explanation points to the investors making rational decisions on the false market perception which appeared rational at the time but proved irrational by correcting their perception in hindsight . The firm declared bankruptcy within 72 hours without any precursor (McDonald & Robinson, 2009), and the only factor that changed during those three days was investors' perception of the company (Housel, 2023; Mawutor, 2014; Fleming & Sarkar, 2014) [1]. Investors made optimal decisions based on this perception, which turned out to be suboptimal once the perception was revealed to be false during those 72 hours (See Appendix B for additional supporting evidence of our hypothesis.).

**Contribution.** We refer to black swan events in stationary environments as **S-BLACK SWAN** and define them in the context of a Markov Decision Process (MDP) as follows:

> **(Informal)** *An* S-BLACK SWAN *event is a state-action pair where humans misperceive both its likelihood and reward. It is perceived as impossible, despite occurring with small probability, while its reward is overestimated relative to its true value in a stationary environment.*

Our work begins with a case study on how S-BLACK SWAN emerge and cause suboptimality gaps in various MDP settings, such as bandit (Theorem 1), small state spaces (Theorem 2), and large state spaces (Theorem 3). We introduced three MDPs to define S-BLACK SWAN : the ground truth MDP (GMDP), the Human MDP (HMDP), and the Human-Estimation MDP (HEMDP). The GMDP represents the real world, while the HMDP reflects humans' biased perceptions (Definitions 1 and 2). S-BLACK SWAN (Definitions 4 and 5) are state-action pairs perceived as impossible in the HMDP but occur with small probability and higher rewards in the GMDP. Our main finding (Theorem 4) shows that while the HEMDP value function asymptotically converges to that of the HMDP over longer horizons, the gap between HMDP and GMDP has a lower bound, influenced by reward distortion, the size of the S-BLACK SWAN set, and their minimum probability of occurrence. Finally, Theorem 5 examines S-BLACK SWAN hitting time, showing that larger reward distortion and higher S-BLACK SWAN probability necessitate more frequent updates to human perception functions.

## 2 PRELIMINARY

**Notations.** The sets of natural, real, nonnegative, and nonpositive real numbers are denoted by $\mathbb{N}$, $\mathbb{R}$, $\mathbb{R}_{\geq 0}$, and $\mathbb{R}_{\leq 0}$ respectively. For a finite set $Z$, the notation $|Z|$ represents its cardinality, and $\Delta(Z)$ denotes the probability simplex on $Z$. Given $X, Y \in \mathbb{N}$ with $X < Y$, we define $[X] := \{1, 2, \ldots, X\}$, the closed interval $[X, Y] := \{X, X+1, \ldots, Y\}$. For $x \in \mathbb{R}_{\geq 0}$, the floor function $\lfloor x \rfloor$ is defined as $\max\{n \in \mathbb{N} \cup \{0\} \mid n \leq x\}$[2].

**Markov Decision Process.** We consider a finite-horizon MDP denoted as $\mathcal{M} = \langle \mathcal{S}, \mathcal{A}, P, R, \gamma, T \rangle$, where $P = \{P_t\}_{t=0}^{T}$ and $R = \{R_t\}_{t=0}^{T}$ for $t \in \mathbb{N}$. Here, $\mathcal{S}$ represents the state space, $\mathcal{A}$ denotes the action space, $P_t : \mathcal{S} \times \mathcal{A} \to \Delta(\mathcal{S})$ is the transition probability function at time $t$, $R_t : \mathcal{S} \times \mathcal{A} \to \mathbb{R}$ is the reward function at time $t$, $\gamma$ is the discount factor, and $T \in \mathbb{N}$ is the horizon length. We define $\mathcal{M}$ as a stationary MDP if $P_t(s' \mid s, a) = P_{t+1}(s' \mid s, a)$ and $R_t(s, a) = R_{t+1}(s, a)$ for all $(s', s, a) \in \mathcal{S} \times \mathcal{S} \times \mathcal{A}$ and for all $t \in [T-1]$. Otherwise, we define $\mathcal{M}$ as a non-stationary MDP. In the stationary case, we denote $P$ and $R$ as the single transition probability function and reward function, respectively. A policy is denoted as $\pi \in \Pi$, where $\Pi : \mathcal{S} \to \Delta(\mathcal{A})$ is the set of policies. We denote a $T$-length trajectory from $\mathcal{M}$ under policy $\pi$ as $\{s_0, a_0, r_0, s_1, a_1, r_1, \ldots, s_{T-1}, a_{T-1}, s_T\}$, where $s_t \sim P_t(\cdot \mid s_{t-1}, a_{t-1})$ and $r_t = R_t(s_t, a_t)$. Assume that all rewards are bounded, i.e., $r_t \in [-R_{\max}, R_{\max}]$ for all $t$. The agent's goal is to compute the optimal

---

[1]The bank's loss endurance, evaluated at 11.7% by the U.S. government, stayed *stationary* over the 72 hours.

[2]For clarity and readability, all notations used throughout the entire paper are elaborated in Appendix A

policy $\pi^\star \in \Pi$ that maximizes the value function: $V_{\mathcal{M}}^\pi(s) \coloneqq \mathbb{E}_\pi[\sum_{t=0}^T \gamma^t R_t(s_t, a_t) \mid P, s_0 = s]$. We further define the normalized visitation probability as $P^\pi(s, a) \coloneqq \frac{1-\gamma^T}{1-\gamma} \sum_{t=0}^{T-1} \gamma^t \mathbb{P}((s_t, a_t) = (s, a) \mid s_0, \pi, P)$, where $\mathbb{P}(s, a \mid s_0, \pi, P)$ is the probability of visiting $(s, a)$ at time $t$ under policy $\pi$ and transition probability $P$ starting from $s_0$ .

The following three theorems, drawn from existing work, lay the groundwork for mathematically formulating *misperception* of the Hypothesis 1.

**Expected Utility Theory.**  Given an outcome space $\mathcal{O} = \{o_1, \ldots, o_K\}$, we define a utility function $g : \mathcal{O} \to \mathbb{R}$ that quantifies the gain or loss associated with each outcome $o_i$. An individual agent is faced with choices, where each choice represents a scenario in which the outcomes $o_i$ occur with given probabilities $p_i$, summing to one. The set of all choices is denoted by $\mathcal{C}$. Each choice $c \in \mathcal{C}$ returns $\mathcal{O}$ with a probability distribution $\boldsymbol{p}_c = (p_1^{(c)}, \ldots, p_K^{(c)})$. Under a given choice $c$, *Expected Utility Theory (EUT)* evaluates the riskiness of that choice as $V(c) = \sum_{i=1}^K g(o_i) p_i^{(c)}$ (von Neumann, 1944; Rabin, 2013). To illustrate, consider a stock market investment scenario where $\mathcal{O} = \{\text{Economic Boom (EB)}, \text{Economic Recession (ER)}\}$. Here, $g(EB)$ represents a gain, while $g(ER)$ represents a loss. The set of choices $\mathcal{C} = \{\text{invest in stocks}, \text{invest in bonds}, \text{keep cash}\}$ corresponds to different probability distributions $\boldsymbol{p}_c = (p_1^{(c)}, p_2^{(c)})$ of outcomes.

**Prospect Theory.**  However, *Expected Utility Theory (EUT)* fails to account for empirical observations from psychological experiments (Drakopoulos & Theodossiou, 2016; Pandit et al., 2019; Wahlberg & Sjoberg, 2000; Vasterman et al., 2005; van der Meer et al., 2022) and economic cases (Rogers, 1998; Wheeler & Wheeler, 2007; BetterUp, 2022) that demonstrate human irrationality. Specifically, humans tend to exhibit internal distortions when perceiving event probabilities $\boldsymbol{p}_c$ and evaluating outcome values $g(\mathcal{O})$ for any choice $c$ (Opaluch & Segerson, 1989). To address these discrepancies, *Prospect Theory (PT)* introduces a probability distortion function $w : [0, 1] \to [0, 1]$ and a value distortion function $u : \mathbb{R} \to \mathbb{R}$, which modify the expected utility calculation to $V(c) = \sum_{i=1}^K u(g(o_i)) w(p_i^{(c)})$ (Kahneman & Tversky, 2013; Fennema & Wakker, 1997). The motivation for introducing *PT* is not only to acknowledge human irrationality but also to provide a more accurate mathematical framework for how people actually perceive probabilities and outcomes. *PT* describes the characteristics of the functions $u$ and $w$ based on empirical case studies. The function $u$ represents *value distortion*, capturing how individuals assess gains and losses ($x$-axis of Figure 1a represents the true value, and the $y$-axis represents the perceived value). The function $w$ represents *probability distortion*, reflecting how individuals tend to overestimate the likelihood of rare events and underestimate the likelihood of more probable events. ($x$-axis of Figure 1b represents the true probability, and the $y$-axis represents the perceived probability.)

**Cumulative Prospect Theory.**  To enhance mathematical rigor—specifically, to ensure that distorted probabilities still sum to one—*Prospect Theory (PT)* was further revised into *Cumulative Prospect Theory (CPT)*. In *CPT*, the expected value is defined as $V(c) = \sum_{i=1}^K u(g(o_i)) \left( w\left(\sum_{j=1}^i p_j^{(c)}\right) - w\left(\sum_{j=1}^{i-1} p_j^{(c)}\right) \right)$, where the function $w$ distorts the cumulative probability of an event $o_i$. The following insurance example illustrates *CPT* in action.

**Example 1** (Insurance policies). *Consider an example where the probability of an insured risk is $1\%$, the potential loss is $1,000$, and the insurance premium is $15$. According to CPT, most would opt to pay the $15$ premium to avoid the larger loss.*

Example 1 shows how a simple decision can be modeled as a two-step Markov Decision Process with states $\mathcal{S} = \{s_{base}, s_{premium}, s_{risk}\}$ representing utility value of $0$, $-15$, and $-1000$, and actions (or choice set $\mathcal{C}$) $\mathcal{A} = \{a_p, a_{np}\}$ for paying or not paying the premium. At $t = 0$, humans choose between $a_p$ (leading to $s_{premium}$) and $a_{np}$, which could result in $s_{base}$ with $99\%$ probability or $s_{risk}$ with $1\%$ probability. Expected

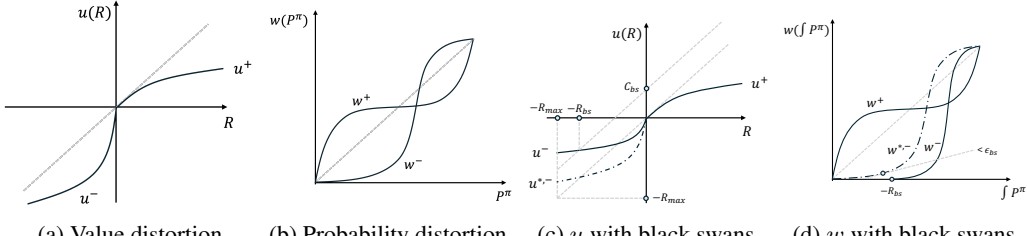

| (a) Value distortion | (b) Probability distortion | (c) $u$ with black swans | (d) $w$ with black swans |

Figure 1: Value distortion function $u$ and probability distortion function $w$. The gray line in Figures 1a and 1b represents $y = x$.

utility theory suggests $a_{np}$ is optimal since its expected value ($V(a_{np}) = -1000 \cdot 0.01 = -10$) is lower than that of $a_p$ ($V(a_p) = -15 \cdot 1 = -15$), but real-world decisions often favor $a_p$, highlighting a divergence from theoretical rationality.

Therefore, we begin by formalizing the key empirical observations from *CPT* into the following definitions.

**Definition 1** (Value Distortion Function). *The value distortion function $u$ is defined as:*

$$u(x) = \begin{cases} u^+(x) & \text{if } x \geq 0, \\ u^-(x) & \text{if } x < 0, \end{cases}$$

*where $u^+ : \mathbb{R}_{\geq 0} \to \mathbb{R}_{\geq 0}$ is non-decreasing, concave with $\lim_{h \to 0^+} (u^+)'(h) \leq 1$, and $u^- : \mathbb{R}_{\leq 0} \to \mathbb{R}_{\leq 0}$ is non-decreasing, convex with $\lim_{h \to 0^-} (u^-)'(h) > 1$.*

**Definition 2** (Probability Distortion Function). *The probability distortion function $w$ is defined as:*

$$w(p_i) = \begin{cases} w^+(p_i) & \text{if } g(x_i) \geq 0, \\ w^-(p_i) & \text{if } g(x_i) < 0, \end{cases}$$

*where $w^+, w^- : [0, 1] \to [0, 1]$ satisfy: $w^+(0) = w^-(0) = 0$, $w^+(1) = w^-(1) = 1$; $w^+(a) = a$ and $w^-(b) = b$ for some $a, b \in (0, 1)$; $(w^+)'(x)$ is decreasing on $[0, a]$ and increasing on $(a, 1]$; $(w^-)'(x)$ is increasing on $[0, b]$ and decreasing on $(b, 1]$.*

The derivative constraints encapsulate the core observations of *CPT*. Specifically, the conditions on $(u^-)'$ and $(u^+)'$ in Definition 1 formalize the tendency for individuals to value losses more heavily than equivalent gains (see Figure 1a). The constraints on $(w^-)'$ and $(w^+)'$ in Definition 2 describe the tendency to overweight (or underweight) the probabilities of rare events and underweight (or overweight) those of average events where the outcome results in a gain (or a loss) (see Figure 1b).

## 3 BLACK SWAN IN STATIONARY AND NON-STATIONARY ENVIRONMENTS

Hypothesis 1 concerns the feasibility of black swan events existing in stationary environments. We next illustrate how black swans can originate from both stationary and non-stationary environments. We begin by defining the black swan event dimension as follows.

**Definition 3** (Black Swan Event Dimension). *For a given MDP $\mathcal{M}$, we define the dimension of a black swan event as the set $\mathcal{S} \times \mathcal{A} \times [T]$.*

Then, we informally refer to $(s, a, t_{bs}) \in \mathcal{S} \times \mathcal{A} \times [T]$ as a black swan event if it represents a rare, high-risk occurrence that significantly deviates from expected outcomes based on prior experience in the real world $\mathcal{M}$. This could involve an unexpected transition or an anomalous reward signal. We then introduce a classification rule that distinguishes black swan events based on whether they occur in non-stationary environments or arise within stationary environments, as follows.

**Algorithm 1** (Black Swan Classification: S-BLACK SWAN ). *For a given (possibly non-stationary) $\mathcal{M}$, suppose $(s, a, t_{bs})$ is a black swan event. If $(s, a, t)$ is a black swan event for $\forall t \in [T]$, then we classify $(s, a, t_{bs})$ as a black swan that originates from environment's stationarity (S-BLACK SWAN ).*

Based on Algorithm 1, one can always identify a unit time interval that classifies any black swan event as an S-BLACK SWAN , as stated in the following proposition.

**Proposition 1.** *If $(s, a, t_{bs})$ is a black swan event, then there exists a time interval $[t_1, t_2] \subseteq [T]$ such that for every $t \in [t_1, t_2]$, the $(s, a, t)$ is classified as S-BLACK SWAN .*

We provide an intuitive interpretation of Proposition 1 through the following example.

**Example 2.** *Suppose $(s, a, t_{bs})$ is a black swan event.*

- *Case 1. Consider $\mathcal{M}$ as a non-stationary MDP where $P_t$ and $R_t$ change at each time step, i.e., $P_t \neq P_{t+1}$ and $R_t \neq R_{t+1}$. If $t_1 = t_2 = t_{bs}$, then $(s, a, t_{bs})$ is classified as an S-BLACK SWAN . However, if $t_1 \neq t_2$ and $t_{bs} \in [t_1, t_2]$, then $(s, a, t_{bs})$ cannot be definitively classified as an S-BLACK SWAN .*
- *Case 2. Consider $\mathcal{M}$ as a piecewise non-stationary MDP where $P_t$ and $R_t$ change every $\lfloor T/k \rfloor$ time steps, i.e., $P_t = P_{t+1}$ and $R_t = R_{t+1}$ for $t \in [kj, kj + (k-1)]$ where $j = 0, 1, \ldots, \lfloor T/k \rfloor$. If $t_1 = kj_{bs}$ and $t_2 = kj_{bs} + (k-1)$, then $(s, a, t_{bs})$ is classified as an S-BLACK SWAN where $j_{bs}$ satisfies $t_{bs} \in [kj_{bs}, kj_{bs} + (k-1)]$.*
- *Case 3. Consider $\mathcal{M}$ as a stationary MDP where $P_t = P_{t+1}$ and $R_t = R_{t+1}$ for all $t \in [T-1]$. In this case, $(s, a, t_{bs})$ is always classified as an S-BLACK SWAN , regardless of the interval $[t_1, t_2]$.*

We then present Case 3 of Example 2 as the following main remark:

**Remark 1.** *If $\mathcal{M}$ is stationary, then any black swan event $(s, a, t)$ is classified as an S-BLACK SWAN . In this case, we omit $t$ and denote the S-BLACK SWAN simply as $(s, a)$.*

Our main goal for the remainder of the paper is to explore Remark 1, with a focus on mathematically defining S-BLACK SWAN within a *stationary* MDP $\mathcal{M}$. We will retain the notation for stationary transition probabilities and reward functions as $P$ and $R$, respectively, omitting the subscript $t$.

## 4 THE EMERGENCE OF S-BLACK SWAN IN SEQUENTIAL DECISION MAKING

We next present a case study to substantiate Hypothesis 1 before formally defining S-BLACK SWAN . We begin by examining how S-BLACK SWANS emerge in sequential decision-making within a *stationary environment*, starting with the bandit case. For a given $(s, a) \in \mathcal{S} \times \mathcal{A}$, let us assume that the function $u$ distorts the reward $R(s, a)$, and the function $w$ distorts the transition probabilities $\{P(s'|s, a)\}_{\forall s' \in \mathcal{S}}$ where $s'$ is the next state. In this Section, we refer to the MDP distorted by functions $u$ and $w$ as the distorted MDP $\mathcal{M}_d := \langle \mathcal{S}, \mathcal{A}, w(P), u(R), \gamma \rangle$, with this notation being used exclusively within this section.

### 4.1 CASE 1. CONTEXTUAL BANDIT ($T = 1$)

We begin with a simple case where the horizon length is $T = 1$, commonly referred to as a contextual bandit (Lattimore & Szepesvári, 2020). Surprisingly, in this setting, the optimal policy of a distorted world coincides with the real world optimal policy as a following Theorem.

**Theorem 1** (One-Step Optimality Deviation). *If $T = 1$, then the optimal policy in the MDP $\mathcal{M}$ is identical to the optimal policy in the distorted MDP $\mathcal{M}_d$.*

Theorem 1 may seem counterintuitive, as Example 1 illustrates that human decision-making often exhibits irrationality. In single-step decision-making, distortions in perception do not significantly affect the optimal policy. For clarification, as shown in Example 1, the perceived reward order remains $u^-(r(s_{loss})) <$

$u^-(r(s_{premium})) < u^-(r(s_{base}))$ because $u^-$ is a non-decreasing convex function. This further implies that a *short* decision horizon may *reduce* the influence of human irrationality.

## 4.2 CASE 2. $|\mathcal{S}| = 2$ WHEN $T > 1$

Now, let us consider the simplest case where $T > 1$ and $|\mathcal{S}| = 2$. Surprisingly, the result that optimality does not deviate still holds similarly to Theorem 1.

**Theorem 2** (Multi-step Optimality Deviation with $|\mathcal{S}| = 2$). *If $|\mathcal{S}| = 2$, then the optimal policy from the MDP $\mathcal{M}$ is also identical to the optimal policy of the distorted MDP $\mathcal{M}_d$ for all $t \in [T]$.*

Theorem 2 may initially seem counterintuitive, given that model errors propagate through distorted transition probabilities and rewards as time $t$ progresses (Janner et al., 2019). However, a straightforward explanation is that for any state-action pair $(s, a) \in \mathcal{S} \times \mathcal{A}$, the function $w$ preserves the order of probabilities. Specifically, if $P(s_1|s, a) > P(s_2|s, a)$, then $w(P(s_1|s, a)) > w(P(s_2|s, a))$ still holds, where $\mathcal{S} = \{s_1, s_2\}$. This suggests that when the state space $|\mathcal{S}|$ is small, the informational complexity required to determine the real-world optimal action remains relatively *low*.

## 4.3 CASE 3. $|S| = 3$ WITH UNBIASED REWARD PERCEPTION

We now consider a general setting with arbitrary $\mathcal{S}$, $\mathcal{A}$, and $T$, but under the assumption that $u(R(s, a)) = R(s, a)$ for all $(s, a)$, indicating that humans have an unbiased perception of their rewards.

**Theorem 3** (Two-step Optimality Deviation with $|\mathcal{S}| = 3$). *If $|\mathcal{S}| = 3$ and $T = 2$, there exists a transition probability function $P$ and a reward function $R$ such that the optimal policy of the MDP $\mathcal{M}$ differs from that of the distorted MDP $\mathcal{M}_d$.*

The optimality deviation in Theorem 3 now aligns with the empirical observation in model-based reinforcement learning; increasing suboptimality is caused by model error propagation (Janner et al., 2019). In summary, Theorems 1, 2, and 3 demonstrate that the discrepancy between the optimal policy derived from human perception and the real-world optimal policy increases as the complexity of the environment ($\mathcal{S}$) grows or as the horizon length ($T$) extends, regardless of the $w$ function.

## 5 AGENT- ENVIRONMENT FRAMEWORK : PERCEPTION AS INTERSECTION

To explore Hypothesis 1, we propose a novel agent-environment framework that treats misperception as information loss in an agent's understanding of the real world [3] (See Figure 2). This framework introduces two *stationary* MDPs: the Human MDP and the Human-Estimation MDP. We begin by defining the *stationary* ground MDP (GMDP) $\mathcal{M}$ as an abstraction of real-world environments without information loss. The following subsections detail the Human MDP (HMDP) and the Human-Estimation MDP (HEMDP).

### 5.1 HUMAN MDP

We define the Human MDP $\mathcal{M}^\dagger = \langle \mathcal{S}, \mathcal{A}, P^\dagger, R^\dagger, \gamma, T \rangle$, where the human (agent) misperceives the visitation probability $P^\pi(s, a)$ through the function $w$, denoted as $P^{\dagger,\pi}(s, a)$, and the reward function $R(s, a)$ through the function $u$, denoted as $R^\dagger(s, a)$. An internal assumption in the HMDP is that its state and action spaces are identical to those of the GMDP $\mathcal{M}$, i.e., $\mathcal{S}^\dagger = \mathcal{S}$ and $\mathcal{A}^\dagger = \mathcal{A}$. Although this assumption may seem unrealistic, especially given that insufficient exploration in large discrete state and action spaces may violate it, the following method shows how the human (agent) can approximate $\mathcal{S}^\dagger$ and $\mathcal{A}^\dagger$ to $\mathcal{S}$ and $\mathcal{A}$, thus supporting this assumption.

---

[3]We detail how misperception reflects information loss from the agent's perspective in Appendix C. .

**Remark 2.** *If the human (agent) cannot perceive a state $s \in \mathcal{S}$, the state space $\mathcal{S}^\dagger$ can be updated to $\mathcal{S}^\dagger \leftarrow \mathcal{S}^\dagger \cup \{s\}$, then set $R^\dagger(s,a) = R(s,a)$ and $P^\dagger(s' \mid s,a) = P(s' \mid s,a)$ while ensuring $P(s \mid s',a) = 0$ for all $s \in \mathcal{S}^\dagger$ and $a \in \mathcal{A}^\dagger$. As a result, the new state $s$ does not influence decision-making in the HMDP, since the probability of the trajectory visiting $s$ remains zero.*

For discrete $\mathcal{S}$ and $\mathcal{A}$, the order statistics of $P^\pi$ can be defined over the sequence $[|\mathcal{S}\|\mathcal{A}|]$, with each $(s,a)$ corresponding to an order index in $[|\mathcal{S}\|\mathcal{A}|]$, enabling the subsequent definition of the cumulative distribution. For brevity, we denote the cumulative distribution of $P^\pi(s,a)$ as $\int P^\pi(s,a)$ . The distortions are then defined by the following relationships:

$$\int P^{\dagger,\pi}(s,a) = \begin{cases} w^+(\int P^\pi(s,a)) & \text{if } R(s,a) \geq 0 \\ w^-(\int P^\pi(s,a)) & \text{if } R(s,a) < 0 \end{cases} , \forall (s,a) \in \mathcal{S} \times \mathcal{A} \tag{1}$$

$$R^\dagger(s,a) = \begin{cases} u^+(R(s,a)) & \text{if } R(s,a) \geq 0 \\ u^-(R(s,a)) & \text{if } R(s,a) < 0 \end{cases} , \forall (s,a) \in \mathcal{S} \times \mathcal{A} \tag{2}$$

We introduce the concept of the *perception gap*: if $\max_{(s,a)} |R(s,a) - R^\dagger(s,a)| < \epsilon_r$, then $R^\dagger(s,a)$ is referred to as an $\epsilon_r$-perceived reward. Similarly, if $\max_{(s,a)} |P^\pi(s,a) - P^{\pi,\dagger}(s,a)| < \epsilon_d$, then $P^{\dagger,\pi}(s,a)$ is called an $\epsilon_d$-perceived visitation probability, where $\epsilon_r, \epsilon_d \in \mathbb{R}_+$. The case where $\epsilon_r = \epsilon_d = 0$ represents an *unbiased perception*. Once the agent perceives $\mathcal{M}$ as $\mathcal{M}^\dagger$, it executes the policy $\pi$ in $\mathcal{M}^\dagger$ and collects a trajectory. Finally, the value function of $\mathcal{M}^\dagger$ is given by $V^\pi_{\mathcal{M}^\dagger}(s) := \mathbb{E}_\pi \left[ \gamma^t R^\dagger(s_t, a_t) \big| P^\dagger, s_0 = s \right]$.

A key challenge in understanding $\mathcal{M}^\dagger$ is why distortions occur in visitation probability rather than transition probability, as discussed in Section 5. This distinction arises because $(s,a)$ is the fundamental event unit (see Remark 1), and a distortion in transition probability implies a distortion in the state itself. The central question, then, is how distortions in visitation probability relate directly to data collection. The following lemma partially addresses this question.

**Lemma 1.** *For a given $\mathcal{M}$, there always exists a function $h : \mathcal{S} \rightarrow \mathcal{S}$ such that $w(\int P^\pi(s,a)) = \int P^\pi(h(s),a)$ holds for any function $w$.*

Our perspective is that distortions in the probability distribution, state space, or other factors lead to distortions in visitation probabilities. With unbiased perception, the agent collects a trajectory $\tau = \{s_0, a_0, r_0, s_1, a_1, \ldots, s_{T-1}, a_{T-1}, s_T\}$. However, when the agent perceives $\mathcal{M}$ as $\mathcal{M}^\dagger$, it observes a perceived trajectory $\tau^\dagger = \{h(s_0), a_0, u(r_0), h(s_1), a_1, \ldots, h(s_{T-1}), a_{T-1}, h(s_T)\}$, where function $h$ distorts the states. Lemma 1 demonstrates that visitation probability distortion arises from state distortion via $h$.

## 5.2 HUMAN-ESTIMATION MDP

After the agent have perceived the ground truth world as $\mathcal{M}^\dagger$, it *estimates* the perceived reward $R^\dagger(s,a)$ as $\widehat{R}^\dagger(s,a)$ and visitation probability $P^{\dagger,\pi}(s,a)$ as $\widehat{P}^{\dagger,\pi}(s,a)$ from its perceived trajectory $\tau^\dagger$. We define a Human-Estimation MDP as $\widehat{\mathcal{M}}^\dagger = \langle \mathcal{S}, \mathcal{A}, \widehat{P}^\dagger, \widehat{R}^\dagger, \gamma, T \rangle$. Note that this estimation process is the same as estimation of generative model in model-based reinforcement learning (Gheshlaghi Azar et al., 2013; Sidford et al., 2018; Agarwal et al., 2020; Kakade, 2003). We also introduce *estimation gap*, that is if $\max_{(s,a)} |R^\dagger(s,a) - \widehat{R}^\dagger(s,a)| \leq \kappa_r$ holds, then $\widehat{R}^\dagger(s,a)$ is $\kappa_r$-estimated reward, and if $\max_{(s,a)} |P^{\pi,\dagger}(s,a) - \widehat{P}^{\pi,\dagger}(s,a)| \leq \kappa_d$ holds, then $\widehat{P}^{\pi,\dagger}(s,a)$ is $\kappa_d$-estimated visitation probability for constant $\kappa_r, \kappa_d \in \mathbb{R}_+$. Finally, the value function of $\widehat{\mathcal{M}}^\dagger$ is given as $V^\pi_{\widehat{\mathcal{M}}^\dagger}(s) := \mathbb{E}_\pi \left[ \gamma^t \widehat{R}^\dagger(s_t, a_t) \big| \widehat{P}^\dagger, s_0 = s \right]$.

$$\text{Environment}$$
$$\mathcal{M} \underset{\epsilon_r, \epsilon_d}{\overset{\text{perception}}{\Longleftrightarrow}} \mathcal{M}^\dagger \underset{\kappa_r, \kappa_d}{\overset{\text{estimation}}{\Longleftrightarrow}} \widehat{\mathcal{M}}^\dagger$$
$$\text{Agent}$$

Figure 2: The agent and environment intersect with perception.

We use the perception and estimation gaps to illustrate the novel agent-environment framework in Figure 2.

## 6 S-BLACK SWAN

Finally, Section 6 provides a definition of S-BLACK SWAN and presents a theoretical analysis aimed at guiding the design of safer ML algorithms in the future.

### 6.1 A DEFINITION OF S-BLACK SWAN

Assume that the rewards for all state-action pairs are ordered as $R_{[1]} \leq \cdots \leq R_{[l]} \leq 0 \leq R_{[l+1]} \leq \cdots \leq R_{[|\mathcal{S}||\mathcal{A}|]}$, and the visitation probabilities are ordered as $P^{\pi}_{[1]} \leq P^{\pi}_{[2]} \leq \cdots \leq P^{\pi}_{[|\mathcal{S}||\mathcal{A}|]}$. We denote the order index of $R(s,a)$ as $I_r(s,a) \in [|\mathcal{S}||\mathcal{A}|]$ and the order index of $P^{\pi}(s,a)$ as $I_p(s,a) \in [|\mathcal{S}||\mathcal{A}|]$, such that $R_{[I_r(s,a)]} = R(s,a)$ and $P^{\pi}_{[I_p(s,a)]} = P^{\pi}(s,a)$. We first provide the definition of S-BLACK SWAN in case of discrete state and action space.

**Definition 4** (S-BLACK SWAN - Discrete State and Action Space). *Given distortion functions $u, w$ and constants $C_{bs} \gg 0$ and $\epsilon_{bs} > 0$, if $(s,a)$ satisfies:*

1. *(High-risk):* $R_{[I_r(s,a)]} - u^-(R_{[I_r(s,a)]}) < -C_{bs}$.
2. *(Rare):* $w^-\left(\sum_{j=1}^{I_p(s,a)} P^{\pi}_{[j]}\right) = w^-\left(\sum_{j=1}^{I_p(s,a)-1} P^{\pi}_{[j]}\right)$, *yet* $0 < P^{\pi}_{[I_p(s,a)]} < \epsilon_{bs}$.

*then we define $(s,a)$ as S-BLACK SWAN .*

Definition 4 finally formalizes the informal concept of black swan events introduced in Section 3. The first property of Definition 4 identifies a *high-risk event* through value distortion. Specifically, if the agent perceives $R$ optimistically, such that $R \ll u^-(R) < 0$, it is classified as a high-risk event (see Figure 1c). The second property characterizes a *rare event* through probability distortion, describing an S-BLACK SWAN event that occurs with a small probability in the real world $\left(0 < P^{\pi}_{[I_p(s,a)]} < \epsilon_{bs}\right)$, but is perceived by the agent as infeasible $\left(w^-\left(\sum_{j=1}^{I_p(s,a)} P^{\pi}_{[j]}\right) = w^-\left(\sum_{j=1}^{I_p(s,a)-1} P^{\pi}_{[j]}\right)\right)$ (See Figure 1d).

The constants $C_{bs}$ and $\epsilon_{bs}$ in Definition 4 quantify the extent of distortion in the functions $u$ and $w$, respectively. Intuitively, $C_{bs}$ and $\epsilon_{bs}$ are directly related to the magnitude of the misperception gap between $\mathcal{M}$ and $\mathcal{M}^{\dagger}$, denoted by $\epsilon_r$ and $\epsilon_p$. This relationship will be further formalized in Theorem 4. We now extend the definition of S-BLACK SWAN to continuous state and action spaces. Suppose the reward function $R : \mathcal{S} \times \mathcal{A} \to \mathbb{R}$ is bijective. Then, the probability $R^{-1} \circ P^{\pi} : \mathbb{R} \to [0,1]$ denotes the probability of a feasible reward induced by policy $\pi$, denoted as $\mathbb{P}_r$. We then have the following definition.

**Definition 5** (S-BLACK SWAN - Continuous State and Action Space). *Given distortion functions $u, w$ and constants $C_{bs} \gg 0$ and $\epsilon_{bs} > 0$, if $(s,a)$ satisfies:*

1. $R(s,a) - u^-(R(s,a)) < -C_{bs}$.
2. $\frac{dw^-(x)}{dx}\big|_{x=F(R(s,a))} \cdot \mathbb{P}_r\big(r = R(s,a)\big) = 0$, *yet* $0 < \mathbb{P}_r\big(r = R(s,a)\big) < \epsilon_{bs}$,

*where $F(r) := \int_{-\infty}^{r} d\mathbb{P}_r$ is the cumulative distribution of $\mathbb{P}_r$, then we define $(s,a)$ as S-BLACK SWAN .*

We then define the minimum probability of S-BLACK SWAN as $\epsilon_{bs}^{\min}$, denoted as $\epsilon_{bs}^{\min} := \min_{(s,a)} \mathbb{P}_r\big(r = R(s,a)\big)$. Let $\mathcal{B}$ denote the collection of all S-BLACK SWAN . For given constants $C_{bs}$ and $\epsilon_{bs}$, we define the distortion functions $w^-$ and $u^-$ that result in $\mathcal{B} = \varnothing$ as $w^-_{\star}$ and $u^-_{\star}$, respectively. Intuitively, $w^-_{\star}$ and $u^-_{\star}$ represent a *safe* perception, meaning that if an agent perceives the world through those, then $\mathcal{B} = \varnothing$. However, it is important to note that $w^-_{\star}$ and $u^-_{\star}$ are not unique functions (see Figure 1d).

## 6.2 THEORETICAL ANALYSIS OF S-BLACK SWAN

Subsection 6.2 explores the properties of S-BLACK SWAN , focusing on how their presence establishes a lower bound on policy performance (Theorem 4) and the timing of their occurrences (Theorem 5), laying the groundwork for future algorithm design. For further analysis, we assume the following.

**Assumption 1** (Relative convexity). *Assume $u_\star^-(r) \le u^-(r)$ holds for $r < 0$.*

Assumption 1 ensures that a human (agent) with $u^-$ perceives rewards more optimistically than one with $u_\star^-$ across all $(s, a)$ pairs. This concept is well illustrated in Figure 1c, where the function $u^-(r) = r$ represents an *unbiased perception*, and deviations from this line indicate increasing reward distortion. In conjunction with Assumption 1, we introduce a proposition regarding S-BLACK SWAN , enabling interpretation within the reward space $[-R_{\max}, R_{\max}]$.

**Proposition 2** (S-BLACK SWAN ). *Let the intersection of the functions $r + C_{bs}$ and $u^-(r)$ occur at $r = -R_{bs}$ (see Figure 1c). Under Assumption 1, if $r(s, a) \in [-R_{\max}, -R_{bs}]$ satisfies:*

1. $r - u^-(r) < -C_{bs}$,
2. $w^-(F(r)) = 0$, with $0 < F(r) < \epsilon_{bs}$,

*then the $(s, a)$ is* S-BLACK SWAN .

A key insight from Proposition 2 is that as $u^-(r)$ approaches $u_\star^-(r)$, the approximation $-R_{bs} \to -R_{\max}$ occurs, finally leading to $|\mathcal{B}| \to 0$ since $|[-R_{\max}, -R_{bs}]| \to 0$ (see Figures 1c). In other words, Proposition 2 demonstrates that reducing the perception gap directly correlates with a decrease in $|\mathcal{B}|$.

Now, to provide an guideline for designing safe learning algorithms to prevent S-BLACK SWAN , it is crucial to quantify how the existence of S-BLACK SWAN leads to an inevitable deviation from the real-world optimal policy. We address this by analyzing how the misperception gap establishes a lower bound on the value function gap between the HMDP $\mathcal{M}^\dagger$ and the GMDP $\mathcal{M}$, as presented in the following theorem.

**Theorem 4** (Convergence of value estimation gap but lower bound on value perception gap). *Under Assumption 1, the asymptotic convergence of the value function estimation holds as follows,*

$$V_{\widehat{\mathcal{M}}^\dagger}^\pi(s) \to V_{\mathcal{M}^\dagger}^\pi(s) \quad a.s. \quad as \quad T \to \infty, \ \forall s, \pi \in \mathcal{S} \times \Pi. \tag{3}$$

*However, under specific conditions on $\epsilon_{bs}, \epsilon_{bs}^{\min}, R_{bs}$, the lower bound of value perception gap as follows.*

$$|V_{\mathcal{M}^\dagger}^\pi(s) - V_{\mathcal{M}}^\pi(s)| = \Omega\left(\frac{\left((R_{\max} - R_{bs})\epsilon_{bs}^{\min} - R_{bs}\epsilon_{bs}\right)(R_{\max} - R_{bs})C_{bs}}{R_{\max}^2}\right) \tag{4}$$

There are two key consequences of Theorem 4. First, Equation (3) demonstrates that the value estimation error converges to zero as the agent rolls out longer trajectories. However, Equation (4) reveals that the value perception gap has a non-zero lower bound, regardless of the horizon length. Equation (4) further indicates that if $u^-(x) \to u_\star^-(x)$ and $w^-(x) \to w_\star^-(x)$, then $R_{bs} \to R_{\max}$ and $\epsilon_{bs} \to 0$ (see Figures 1c and 1d), leading to the convergence of this lower bound to zero. Second, Equation (4) aligns with the intuition that greater distortion in reward perception (i.e., larger $C_{bs}$) and an increased number of S-BLACK SWAN (i.e., larger $(R_{\max} - R_{bs})$) coupled with a higher minimum probability of S-BLACK SWAN occurrence (i.e., larger $\epsilon_{bs}^{\min}$) result in a higher lower bound. Therefore, Theorem 4 concludes that even with zero estimation error, a lower bound on approximating the true value function remains, and this lower bound increases as $C_{bs}$ and $\epsilon_{bs}^{\min}$ become more pronounced.

Then, the next natural question is *how to decrease that lower bound*, specifically, how can an agent can learn to self-correct toward a safe perception, i.e., $u^- \to u_\star^-$ and $w^- \to w_\star^-$. This question can be further refined to: *What is the probability of encountering* S-BLACK SWAN *if the agent takes $t$ steps?* We address this under the assumption of non-zero one-step reachability, as follows.

**Theorem 5** (S-BLACK SWAN hitting time). *Assume* $\mathbb{P}_{\pi^*}(s' \mid s) > 0$ *for any* $s, s' \in \mathcal{S}$, *indicating that the one-step state reachability equipped with optimal policy is non-zero, and consider that one step corresponds to a unit time. Then, if the agent takes* $t$ *steps such that* $t \geq \log\left(\frac{\delta}{p_{\min}}\right) / \log(1 - p_{\max}) + 1$, *where* $p_{\min} = \frac{R_{\max} - R_{bs}}{2R_{\max}} \epsilon_{bs}^{\min}$ *and* $p_{\max} = \frac{R_{\max} - R_{bs}}{2R_{\max}} \epsilon_{bs}$, *it will encounter* S-BLACK SWAN *with at least probability* $\delta \in (0, 1]$.

A key takeaway of Theorem 5 is determining how often a human should correct their internal perception. A large perception gap ($R_{\max} - R_{bs}$) and frequent occurrence of black swan events ($\epsilon_{bs}^{\min}$) require more frequent execution of the self-perception correction algorithm.

## 7    RELATED WORKS: NECESSITY OF S-BLACK SWAN

This section discusses safe reinforcement learning (RL) algorithms, emphasizing the limitations of existing approaches in addressing black swan events and highlighting the need for a new perspective[4].

Safe RL algorithms are generally classified into three approaches: worst-case criterion, risk-sensitive criterion, and constrained criterion (García & Fernández, 2015). However, these approaches face significant limitations when dealing with black swan events. The worst-case criterion, which optimizes policy performance under the least favorable scenarios by maximizing the minimum return, becomes overly conservative when black swan events are considered, as they expand the uncertainty set $\mathcal{W}$, leading to impractical decisions such as avoiding all risky activities or adopting extreme safety measures (Heger, 1994; Coraluppi, 1997; Coraluppi & Marcus, 1999; 2000). Similarly, risk-sensitive algorithms, which incorporate a sensitivity factor to balance return maximization and risk management (Howard & Matheson, 1972; Chung & Sobel, 1987; Patek, 2001), are inadequate for handling black swan events because return variance, a commonly used risk measure, fails to account for the fat tails in distributions (Huisman et al., 1998; Bradley & Taqqu, 2003; Bubeck et al., 2013; Agrawal et al., 2021). Additionally, log-exponential utility functions, often associated with robust MDPs, do not effectively address the risks posed by black swans (Osogami, 2012; Moldovan & Abbeel, 2012; Leqi et al., 2019). The constrained criterion, which maximizes expected returns while meeting multiple utility constraints such as return variance or minimum thresholds (Geibel, 2006; Delage & Mannor, 2010; Ponda et al., 2013; Di Castro et al., 2012), also faces challenges with black swan events. These events complicate threshold selection, often necessitating more conservative policies, and suggest that constraints should be redefined to focus on state and action-specific risks rather than overall returns (Bagnell et al., 2001; Iyengar, 2005; Nilim & El Ghaoui, 2005; Wiesemann et al., 2013; Xu & Mannor, 2010). Furthermore, distributional RL is vulnerable to black swans, as extreme outliers in the reward distribution slow the convergence of the Bellman operator and provide a large suboptimality gap due to biased return expectations (Bellemare et al., 2017).

In summary, traditional risk criteria in RL are insufficient for managing the unique risks associated with black swan events, highlighting the need for novel approaches.

## 8    CONCLUSION

In conclusion, this paper redefines black swan events by introducing S-BLACK SWAN , highlighting that such high-risk, rare events can occur even in unchanging environments due to human misperception. We categorized and mathematically formalized these events, aiming to guide the development of algorithms that correct human perception to prevent such occurrences. This work opens the door for future research to enhance decision-making systems and reduce the impact of black swan events.

---

[4]Further details are in Appendix D, along with a discussion of CPT's application in risk analysis in Appendix E.

ACKNOWLEDGMENTS

We would like to express our heartfelt gratitude to Jason Jangho Choi for his thoughtful and instructive insights on the early draft of this paper. Special thanks to Donghao Ying for the enriching discussions on mathematical notation. Our sincere appreciation goes to Theophane Weber, Csaba Szepesvari, and the rest of Google DeepMind team for their careful review of the first draft. We are also grateful to John D Martin from the RLC 2024 conference for the engaging discussions on algorithm design to prevent black swan events. We would also like to thank Negar Mehr, Mark Muller, Panayiotis Papadopoulos, and all the participants of the Berkeley control seminar for their invaluable and insightful contributions during our discussions of the second draft (ICLR submitted paper). Lastly, we extend our warmest thanks to the BAIR researchers at UC Berkeley, including Suhong Moon for the kind invitation to speak in John Canny's lab, and Seohong Park for providing such thorough and constructive comments on the second draft.

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

## A  NOTATIONS

This section provides a table summarizing all the notations and their meanings introduced in the main paper.

| Notation | Meaning | Defintion |
|---|---|---|
| $\mathcal{M}$ | Ground truth MDP | Section 2 |
| $\mathcal{S}$ | State space | Section 2 |
| $\mathcal{A}$ | Action space | Section 2 |
| $T$ | Time horizon | Section 2 |
| $P$ | Transition probability function of $\mathcal{M}$ | Section 2 |
| $R$ | Reward function of $\mathcal{M}$ | Section 2 |
| $\gamma$ | Discount factor | Section 2 |
| $\pi$ | Policy | Section 2 |
| $P^\pi$ | Normalized visitation probability of $\mathcal{M}$ | Section 2 |
| $V_\mathcal{M}^\pi$ | Value function of $\mathcal{M}$ | Section 2 |
| $u, u^-, u^+$ | Value distortion function | Section 2 (Figure 1a) |
| $w, w^-, w^+$ | Probability distortion function | Section 2 (Figure 1b) |
| $\mathcal{M}_d$ | Distorted MDP | Section 4 |
| $w(P)$ | Transition probability of $\mathcal{M}_d$ | Section 4 |
| $u(R)$ | Reward function of $\mathcal{M}_d$ | Section 4 |
| $\mathcal{M}^\dagger$ | Human MDP (HMDP) | Subsection 5.1 |
| $P^\dagger$ | Transition probability of $\mathcal{M}^\dagger$ | Subsection 5.1 |
| $R^\dagger$ | Reward function of $\mathcal{M}^\dagger$ | Subsection 5.1 |
| $P^{\dagger,\pi}$ | Normalized visitation probability of $\mathcal{M}^\dagger$ | Subsection 5.1 |
| $\int P^\pi$ | Cumulative visitation distribution of $\mathcal{M}$ | Subsection 5.1 |
| $\int P^{\dagger,\pi}$ | Cumulative visitation distribution of $\mathcal{M}^\dagger$ | Subsection 5.1 |
| $V_{\mathcal{M}^\dagger}^\pi$ | Value function of $\mathcal{M}^\dagger$ | Subsection 5.1 |
| $\epsilon_r, \epsilon_d$ | Perception gap | Subsection 5.1 |
| $\widehat{\mathcal{M}}^\dagger$ | Human Estimation MDP (HEMDP) | Subsection 5.2 |
| $\widehat{P}^\dagger$ | Transition probability of $\widehat{\mathcal{M}}^\dagger$ | Subsection 5.2 |
| $\widehat{R}^\dagger$ | Reward function of $\widehat{\mathcal{M}}^\dagger$ | Subsection 5.2 |
| $V_{\widehat{\mathcal{M}}^\dagger}^\pi$ | Value function of $\widehat{\mathcal{M}}^\dagger$ | Subsection 5.2 |
| $\kappa_r, \kappa_d$ | Estimation gap | Subsection 5.2 |
| $R_{[\cdot]}, P_{[\cdot]}^\pi$ | Order statistics of reward and visitation probability | Section 6 |
| $\mathbb{P}_r$ | Probability of reward | Section 6 |
| $F(r)$ | Cumulative distribution function of $\mathbb{P}_r$ | Section 6 |
| $\mathcal{B}$ | Collection of all S-BLACK SWAN | Section 6 |

| Notation | Meaning | Defintion |
|:---:|:---:|:---:|
| $C_{bs}, \epsilon_{bs}$ | The extent of distortion of functions $u$ and $w$ | Section 6 (Figures 1c and 1d) |
| $\epsilon_{bs}^{\min}$ | Minimum probability of S-BLACK SWAN | Section 6 |
| $u_\star^-$ | $u^-$ that satisfies $\mathcal{B} = \varnothing$, i.e. safe reward perception | Section 6 |
| $w_\star^-$ | $w^-$ that satisfies $\mathcal{B} = \varnothing$, i.e. safe probability perception | Section 6 |

## B  SUPPORTING EVIDENCE

This section provides supporting evidence for this paper's main perspective; how interpreting black swans through the lens of human misperception is meaningful and applicable in the real-world scenarios beside Lehman Brothers Bankruptcy case introduced in Introduction.

**Case1. Unexpected Drowning of NASA Astronauts Due to Overlooked Details**   (The following is a summary of the case; please refer to (Clark, 2021; Brinkley, 2019) for a more detailed account.) Before launching rockets, NASA conducted tests on its space suits using high-altitude hot-air balloons. On May 4, 1961, Victor Prather and another pilot ascended to 113,720 feet to evaluate the suit's performance. While the test itself was successful, an unforeseen risk led to tragedy during the planned ocean landing. Prather opened his helmet faceplate to breathe fresh air, and as he slipped into the water while attaching to a rescue line, his now-exposed suit filled with water. Despite NASA's rigorous planning and preparation, the risk of opening the faceplate - perceived as an extremely minor detail - was underestimated, resulting in catastrophic consequences. This highlights how rigorously meticulous planning can still fail to account for overlooked events.

**Case2. Healthcare**   Diabetes patients typically experience a highly chronic condition, making their state relatively predictable a few hours into the future (stationary environment). However, rare hypoglycemic events, characterized by a sudden and dangerous drop in blood sugar, pose significant risks. To address this, Wang et al. (2023) developed an RL model capable of predicting changes in a patient's condition and providing optimized treatments. Furthermore, (Panda et al., 2020; Ambhika et al., 2024) emphasize the critical importance of selecting appropriate signals as inputs, as human misperceptions about what constitutes important signals can lead to unexpected and suboptimal decisions. As a result, traditional supervised learning methods, such as general Transformers paired with loss functions like MSE, may struggle to accurately predict rare events like hypoglycemic episodes when the input signals fail to align with the underlying dynamics.

## C  MISPERCEPTION IS INFORMATION LOSS

Based on Hypothesis 1, this prompts us to investigate the concept of *misperception*. Initially, we must clearly define what constitutes *perception*. In *The Quest for a Common Model of the Intelligent Decision Maker*, Sutton defines perception as one of four principal components of agents, stating: "The perception component processes the stream of observations and actions to produce the subjective state, a summary of the agent-world interaction so far that is useful for selecting action (the reactive policy), for predicting future reward (the value function), and for predicting future subjective states (the transition model)" (Sutton, 2022). This definition leads us to consider misperception as the *information loss* occurring when processing observations into the subjective state, such that the reward and transition model are not equivalent to those from the environment. The interpretation of misperception as *information loss during processing* is somewhat ambiguous, depending on how the boundary between the agent and the environment is defined. Turing first proposed the

concept of a boundary between the agent and environment as a 'skin of an onion' (Turing, 2009), and later, Jiang (2019) suggested that algorithms are not boundary-invariant.

Therefore, we propose a new agent-environment framework that incorporates the notion that *misperception is the information loss from an agent's processing*. This framework positions perception at the intersection between the agent and the environment. We provide a detailed description of our agent-environment framework in Figure 2.

# D   RELATED WORKS: NECESSITY OF A NEW PERSPECTIVE TO UNDERSTAND BLACK SWANS AND EVIDENCE FOR HYPOTHESIS 1

In this section, we focus not only on addressing the necessity of a new perspective to understand black swan events but also on providing evidence for the proposed perspective of black swan origin (Hypothesis 1). This is concretized by examining the following two questions. First, in Subsection D.1, we discuss the insufficiency of existing decision-making rules under risk by exploring related works, which support the need for a new perspective to understand black swans. Specifically, we address *why existing safe reinforcement learning strategies for solving Markov Decision Processes are insufficient to handle black swan events?*. If this premise is validated, then in Subsection D.2, we elaborate on the motivation and related works that support our informal hypothesis of black swan origin (Hypothesis 1). Specifically, we explore *how irrationality relates to misperception and how irrationality could bring about black swan events*.

## D.1   DECISION MAKING UNDER RISK

Based on the comprehensive survey on safe reinforcement learning in Garcıa & Fernández (2015), the algorithms can be classified into threefold: worst case criterion, risk-sensitive criterion and constraint criterion. We elaborate on why the existence of black swans in the environment renders these three approaches insufficient.

**Worst case criterion.** Learning algorithms of the worst case criterion focus on devising a control policy that maximizes policy performance under the least favorable scenario encountered during the learning process, defined as $\max_{\pi \in \Pi} \min_{w \in \mathcal{W}} V_{\mathcal{M}}^{\pi}(s; w)$, where $\mathcal{W}$ represents the set of uncertainties. This criterion can be categorized based on whether $\mathcal{W}$ is defined in the environment or in the estimation of the model. The presence of black swan events in the worst case, where $\mathcal{W}$ represents aleatoric uncertainty of the environment (Heger, 1994; Coraluppi, 1997; Coraluppi & Marcus, 1999; 2000), results in overly conservative, and thus potentially ineffective, policies. This occurs because the significant impact of black swan events inflates the size of $\mathcal{W}$, even though such events are rare. In practical terms, this could manifest itself as abstaining from any economic activity ($\pi$), such as not investing in stocks or not depositing a check against future potential bankruptcies ($\min_{w \in \mathcal{W}} V_{\mathcal{M}}^{\pi}(s; w)$) in order to maximize its income ($\max_{\pi \in \Pi}(\triangle)$), or maintaining constant health precautions such as wearing mask or maintaining distance with groups ($\pi$) to prepare for a possible pandemic ($\triangle = \min_{w \in \mathcal{W}} V_{\mathcal{M}}^{\pi}(s; w)$) in order to maintain its health ($\max_{\pi \in \Pi}$). Similarly, when $\mathcal{W}$ encompasses the uncertainty of the model parameter (Bagnell et al., 2001; Iyengar, 2005; Nilim & El Ghaoui, 2005; Wiesemann et al., 2013; Xu & Mannor, 2010) - as seen in robust MDP or distributionally robust MDP - this aligns closely with our black swan hypothesis, where misperception of the world model is similar to uncertainty in model estimation. However, the need to accommodate black swan events requires enlarging the possible set of models ($|\mathcal{W}|$), leading to extremely conservative policies. This can be likened to performing an overly pessimistic portfolio optimization ($\pi$), where every bank is assumed to have a minimal but possible risk of bankruptcy ($\min_{w in \mathcal{W}} V_{\mathcal{M}}^{\pi}(s; w)$), thus influencing asset allocation strategies ($\max_{\pi \in \Pi} \min_{w \in \mathcal{W}} V_{\mathcal{M}}^{\pi}(s; w)$) to be extremely conservative in asset investing.

**Risk sensitive criterion.** Risk-sensitive algorithms strike a balance between maximizing reinforcement and mitigating risk events by incorporating a sensitivity factor $\beta < 0$ (Howard & Matheson, 1972;

Chung & Sobel, 1987; Patek, 2001). These algorithms optimize an alternative value function $V_{\mathcal{M}}^{\pi}(s) = \beta^{-1} \log \mathbb{E}_{\pi}[\exp^{\beta G} | P, s_0 = s]$, where $\beta$ controls the desired level of risk and $G := \sum_{t=0}^{T} \gamma^t R(s_t, a_t)$ is a cumulative return. However, it is recognized that associating risk with the variance of the return is practical, as in $V_{\mathcal{M}}^{\pi}(s) = \beta^{-1} \log \mathbb{E}_{\pi}[\exp^{\beta G}] = \max_{\pi \in \Pi} \mathbb{E}_{\pi}[G] + \frac{\beta}{2} \text{var}(G) + \mathcal{O}(\beta^2)$, and the existence of black swan events does not significantly affect the returns of variance $(\text{var}(G))$ due to their rare nature. It should be noted that risk-sensitive approaches are not well suited for handling black swan events, as the same policy performance with small variance can entail substantial risks (Geibel & Wysotzki, 2005). More generally, the objective of the exponential utility function is one example of risk-sensitive learning based on a trade-off between return and risk, i.e., $\max_{\pi \in \Pi}(\mathbb{E}_{\pi}[G] - \beta w)$ (Zhang et al., 2018), where $w$ is replaced by $\text{Var}(G)$. This approach is known in the literature as the variance-penalized criterion (Gosavi, 2009), the expected value-variance criterion (Taha, 2007; Heger, 1994), and the expected-value-minus-variance criterion (Geibel & Wysotzki, 2005). However, a fundamental limitation of using return variance as a risk measure is that it does not account for the fat tails of the distribution (Huisman et al., 1998; Bradley & Taqqu, 2003; Bubeck et al., 2013; Agrawal et al., 2021). Consequently, risk can be underestimated due to the oversight of low probability but highly severe events (black swans).

Furthermore, a critical question arises regarding whether the log-exponential function belongs to *appropriate utility function class* for defining *real-world risk*. Risk-sensitive MDPs have been shown to be equivalent to robust MDPs that focus on maximizing the worst-case criterion, indicating that the log-exponential utility function may not be beneficial in the presence of black swans (Osogami, 2012; Moldovan & Abbeel, 2012; Leqi et al., 2019). This issue was first raised by Leqi et al. (2019) and led to the proposal of a more realistic risk definition called 'Human-aligned risk', which also incorporates human misperception akin to our informal black swan hypothesis (Hypothesis 1).

**Constrained Criterion.** The constrained criterion is applied in the literature to constrained Markov processes where the goal is to maximize the expected return while maintaining other types of expected utilities below certain thresholds. This can be formulated as $\max_{\pi \in \Pi} \mathbb{E}_{\pi}[G]$ subject to $N$ multiple constraints $h_i(G) \leq \alpha_i$, for $i \in [N]$, where $h_i : \mathbb{R} \to \mathbb{R}$ is a function of return $G := \sum_{t=0}^{T} \gamma^t R(s_t, a_t)$ (Geibel, 2006). Typical constraints include ensuring the expectation of return exceeds a specific minimum threshold ($\alpha$), such as $\mathbb{E}_{\pi}[G] \geq \alpha$, or softening these hard constraints by allowing a permissible probability of violation ($\epsilon$), such as $\mathbb{P}(\mathbb{E}_{\pi}[G] \geq \alpha) \geq 1 - \epsilon$, known as chance-constraint (Delage & Mannor (2010); Ponda et al. (2013)). Constraints might also limit the return variance, such as $\text{Var}(G) \leq \alpha$ (Di Castro et al. (2012)). However, the presence of black swans highlights one of the challenges with the Constrained Criterion, specifically the appropriate selection of $\alpha$. The presence of black swans necessitates a lower $\alpha$, which in turn leads to more conservative policies. Furthermore, a black swan event is determined at least by the environment's state and its action, rather than its full return. Therefore, constraints should be redefined over more fine-grained inputs—not merely returns, but in terms of state and action—which leads to our definition of black swan dimensions (Definition 3).

## D.2 How irrationality relates with spatial black swans.

Before starting Subsection D.2, we clarify that the term *irrationality* is used here to denote rational behavior based on a false belief. In this subsection, we first review existing work on the four rational axioms and then claim how two of these axioms should be modified to account for *irrationality* in human decision-making.

**Rationality in decision making.** In the foundation of decision theory, rationality is understood as internal consistency (Sugden (1991); Savage (1972)). A prerequisite for achieving rationality in decision-making is the ability to compare outcomes, denoted as set $\Omega$ where $|\Omega| = N$, through a *preference* relation in a *rational* manner. In von Neumann (1944), it is demonstrated that preferences, combined with *rationality axioms* and probabilities for possible outcomes, denoted as $p_i$ which is a probability of outcome $o_i \in \Omega$, imply the existence of utility values for those outcomes that express a preference relation as the expectation

of a scalar-valued function of outcomes. Define the choice (or lotteries) as set $\mathcal{L}$, which is a combination of selecting total $N$ outcomes, that is, $\sum_{i=1}^{N} p_i o_i$. The essential rationality axioms are as follows.

1. Completeness: Given two choices, either one is preferred over the other or they are considered equally preferable.
2. Transitivity: If $A$ is preferred to $B$ and $B$ is preferred to $C$, then $A$ must be preferred to $C$.
3. Independence: If $A$ is preferred to $B$, and a event probability $p \in [0, 1]$, then $pA + (1 - p)C$ should be preferred to $pB + (1 - p)C$.
4. Continuity: If $A$ is preferred to $B$ and $B$ is preferred to $C$, there exists a event probability $p \in [0, 1]$ such that $B$ is considered equally preferable to $pA + (1 - p)C$.

Expanding on these axioms, Sunehag & Hutter (2015) extends rational choice theory to encompass the full reinforcement learning problem, further axiomatizing the concept in Sunehag & Hutter (2011) to establish a rational reinforcement learning framework that facilitates optimism, crucial for systematic explorative behavior. Subsequent studies focusing on defining rationality in reinforcement learning, such as Shakerinava & Ravanbakhsh (2022); Bowling et al. (2023), concentrate on the axioms of assigning utilities to all finite trajectories of a Markov Decision Process. Specifically, Shakerinava & Ravanbakhsh (2022); Bowling et al. (2023) clarify the reward hypothesis Sutton that underpins the design of rational agents by introducing an additional axiom to existing rationality axioms. Furthermore, Pitis (2024) explores the design of multi-objective rational agents, and Carr et al. (2024) explores and defines rational feedback in Large Language Models (LLMs) by investigating the existence of optimal policies within a framework of learning from rational preference feedback (LRPF).

**Irrationality due to subjective probability.** The definition of irrationality and its origins has been extensively investigated through case studies in various fields such as psychology, education, and particularly economics. Simon (1993) defined irrationality as being poorly adapted to human goals, diverging from the norm of human's object, influenced by emotional or psychological factors in decision-making. Subsequently, Martino et al. (2006); Gilovich et al. (2002) further concretized what exactly these *emotional or psychological factors* entail by describing them as information loss during human perception of the real world. More specifically, Martino et al. (2006) pointed out that in a world filled with symbolic artifacts, where optimal decision-making often requires skills of abstraction and decontextualization, such mechanisms may render human choices irrational. Further studies, such as Opaluch & Segerson (1989), scrutinize more deeply and classify the *irrationality* of human behavior into five factors: subjective probability, regret/disappointment, reference points, complexity, and ambivalence.

In this paper, we focus on the *subjective probability* factor to elucidate the relationship between irrationality and spatial black swans. Opaluch & Segerson (1989) explores subjective probabilities as an early modification to the expected utility model from von Neumann (1944), focusing on decision-makers who rely on *personal beliefs* about probabilities rather than objective truths. This minor conceptual shift can lead to significant behavioral changes due to the imperfect information and processing abilities of individuals. Especially, Opaluch & Segerson (1989) highlights the difficulty in accurately estimating the probability of *rare events* - such as black swans - which often leads to critical errors in judgment. These errors occur because rare events provide insufficient data for accurate probability estimation or are misunderstood due to their infrequency, leading to perceptions that such events are either less likely or virtually impossible. This misperception is exemplified in various scenarios, such as:

1. An individual working in a dangerous job who has never personally observed an accident may underestimate the probability of an accident occurring Drakopoulos & Theodossiou (2016); Pandit et al. (2019).
2. Media coverage of events such as plane crashes may cause an overestimation of the probability of a crash, since the public is aware of all crashes but not of all safe trips Wahlberg & Sjoberg (2000); Vasterman et al. (2005); van der Meer et al. (2022).

3. The popularity of purchasing lottery tickets may be explainable in terms of people's inability to comprehend the true probability of winning, influenced instead by news accounts of 'real' people who win multi-million dollar prizes (Rogers (1998); Wheeler & Wheeler (2007); BetterUp (2022)).

## E   CUMULATIVE PROSPECT THEOREM AND RISK

We note that existing works on incorporating cumulative prospect theory (CPT) into reinforcement learning, such as (Prashanth et al. (2016); Jie et al. (2018); Danis et al. (2023)), primarily focus on estimating the CPT-based value function and optimizing it to derive an optimal policy. Specifically, (Prashanth et al. (2016); Jie et al. (2018)) demonstrate how to estimate the CPT value function using the Simultaneous Perturbation Stochastic Approximation method and how to compute its gradient for policy optimization algorithms. Additionally, (Shen et al. (2014); Ratliff & Mazumdar (2019)) proposed a novel Q-learning algorithm that applies a utility function to Temporal Difference (TD) errors and demonstrated its convergence. However, these studies (Prashanth et al. (2016); Jie et al. (2018); Danis et al. (2023); Shen et al. (2014); Ratliff & Mazumdar (2019)) do not focus on learning the utility and weight functions, $u$ and $w$, but rather assume these as simple functions and focus on how to *estimate* these functions.

However, this study aims to elucidate the mechanisms by which black swan events arise from the discrepancies between $\mathcal{M}^{\dagger}$ and $\mathcal{M}$, despite the agent having perfect estimation, i.e., $\kappa_r = 0, \kappa_p = 0$. As future work, concentrating on devising strategies to *reweight* the functions $u^+, u^-$, and $w$ to mitigate the divergence between the Human MDP $\mathcal{M}^{\dagger}$ and the ground truth MDP $\mathcal{M}$ is suggested as a way to achieve antifragility.

## F   PRELIMINARY FOR PROOFS

This subsection covers the preliminary concepts necessary for proving the theorems and lemmas presented in the paper.

First, in a discrete state and action space, the value function $\mathcal{M}$ could be expressed as an inner product of reward function $R$ and normalized occupancy measure $P^{\pi}$ as follows,

$$V_{\mathcal{M}}(s_0) = \frac{1-\gamma^T}{1-\gamma} \sum_{(s,a)\in\mathcal{S}\times\mathcal{A}} R(s,a)P^{\pi}(s,a) \tag{5}$$

Based on Equations (5), (1), and (2), the *CPT* distorts the reward and its visitation probability as follows,

$$V_{\mathcal{M}^{\dagger}}(s_0) = \frac{1-\gamma^T}{1-\gamma} \sum_{s,a\in\mathcal{S}\times\mathcal{A}} u(R(s,a))\frac{d}{dsda}w\left(\int P^{\pi}(s,a)\right). \tag{6}$$

where $\dagger$ denotes the value function that was distorted due to misperception. As one property of CPT is that human perception exhibits distinct distortions of events based on whether the associated rewards are positive or negative, we divide the functions $u(R(s,a))$ and $w(\int P^{\pi}(s,a))$ into $u^-(R(s,a)), w^-(\int P^{\pi}(s,a))$ where $R(s,a) < 0$, and $u^+(R(s,a)), w^+(\int P^{\pi}(s,a))$ where $R(s,a) \geq 0$. Assume that the rewards from all state-action pairs $R(s,a)$ are ordered as $R_{[1]} \leq \cdots \leq R_{[l]} \leq 0 \leq R_{[l+1]} \leq \cdots \leq R_{[|\mathcal{S}\|\mathcal{A}|]}$, and the visitation probability as $P^{\pi}_{[1]} \leq P^{\pi}_{[2]} \leq \cdots \leq P^{\pi}_{[|\mathcal{S}\|\mathcal{A}|]}$. Then, the Equation (6) can be represented as follows:

$$V_{\mathcal{M}^\dagger}(s_0) = \frac{1-\gamma^T}{1-\gamma}\left(\sum_{i=1}^{|\mathcal{S}||\mathcal{A}|} u(R_{[i]})\left(w\left(\sum_{j=1}^{i} P^\pi_{[j]}\right) - w\left(\sum_{j=1}^{i-1} P^\pi_{[j]}\right)\right)\right)$$

$$= \sum_{i=1}^{l} u^-(R_{[i]})\left(w^-\left(\sum_{j=1}^{i} P^\pi_{[j]}\right) - w^-\left(\sum_{j=1}^{i-1} P^\pi_{[j]}\right)\right)$$

$$+ \sum_{i=l+1}^{|\mathcal{S}||\mathcal{A}|} u^+(R_{[i]})\left(w^+\left(\sum_{j=i}^{|\mathcal{S}||\mathcal{A}|} P^\pi_{[j]}\right) - w^+\left(\sum_{j=i+1}^{|\mathcal{S}||\mathcal{A}|} P^\pi_{[j]}\right)\right) \tag{7}$$

If we define the reward as the random variable $X$, then we can regard its instance as $R_{[i]}$ and its probability as $P^\pi_{[i]}$ where the probability is dependent on the policy $\pi$. Suppose that reward function $R : \mathcal{S} \times \mathcal{A} \to \mathbb{R}$ is one to one function. Then the probability $R^{-1} \circ P^\pi : \mathbb{R} \to [0,1]$ denotes the probability of reward and we denote it as $\mathbb{P}_r$. Then, for a reward random variable $\mathcal{R} \sim \mathbb{P}_r$, expanding the how CPT- applied value function look like in Equation (4), we can rewrite the Equation (7) based on continuous state and actions space as follows.

$$V_{\mathcal{M}^\dagger}(s_0) = \int_0^\infty w^+\left(\mathbb{P}_r(u^+(\mathcal{R}) > r)\right) dr - \int_0^\infty w^-\left(\mathbb{P}_r(u^-(\mathcal{R}) > r)\right) dr \tag{8}$$

We use the fact that for real-value function $g$, it holds that $\mathbb{E}[g(\mathcal{R})] = \int_0^\infty \Pr(g(\mathcal{R}) > r)dr$. Within the above problem setting, the agent's goal is to estimate the value function under safe perception $u^-_\star, w^-_\star$ as follows:

$$V_{\mathcal{M}}(s_0) = \int_0^\infty w^+\left(\mathbb{P}_r(u^+(X) > r)\right) dr - \int_0^\infty \boldsymbol{w}^-_\star\left(\mathbb{P}_r(\boldsymbol{u}^-_\star(X) > r)\right) dr \tag{9}$$

Note that the safe perception is only defined over $w^-$ and $u^-$ as $w^-_\star$ and $u^-_\star$. However, the agent possesses its own perceptions $\mathcal{M}^\dagger$, for which we assume the risk perception is represented as:

$$V_{\mathcal{M}^\dagger}(s_0) = \int_0^\infty w^+\left(\mathbb{P}_r(u^+(X) > r)\right) dr - \int_0^\infty \boldsymbol{w}^-\left(\mathbb{P}_r(\boldsymbol{u}^-(X) > r)\right) dr \tag{10}$$

As time goes by, the agent's goal is approximating the weight functions and utility functions such as $w^- \to w^-_\star$ and $u^- \to u^-_\star$. Then, by the single trajectory data up to time $t$, i.e. $\{h(s_i), a_i, u(r_i), h(s_{i+1})\}_{i=0}^t$ where the reward value itself and its sampling distribution are distorted due to the functions $u$ and $w$, respectively (see Lemma 1 for definition of function $h$). Since function $h$ maps state space to state space, we just use the notation $\{s'_i, a_i, u(r_i), s'_{i+1}\}_{i=0}^t$ to denote Let $r_i, i = 1, .., t$ denote $n$ samples of the reward random variable $X$. We define the empirical distribution function (EDF) for $u^+(X)$ and $u^-(X)$ as follows

$$\hat{F}^+_t(r) = \frac{1}{t}\sum_{i=1}^n \mathbf{1}_{(u^+(r_i) \le r)}, \quad \text{and} \quad \hat{F}^-_t(r) = \frac{1}{t}\sum_{i=1}^n \mathbf{1}_{(u^-(r_i) \le r)}.$$

Using the EDFs, the CPT value up to time $t$ can be estimated as follows,

$$V_{\widehat{\mathcal{M}^\dagger}}(s_0) = \int_0^\infty w^+\left(1 - \hat{F}^+_t(r)\right) dr - \int_0^\infty w^-\left(1 - \hat{F}^-_t(r)\right) dr \tag{11}$$

Again, we note that the gap between $\mathcal{M}$ and $\mathcal{M}^\dagger$ is defined over a gap between $(u^-, w^-)$ and $(u^-_\star, w^-_\star)$ that is proportional to the existence of spatial black swan events.

## G  PROOFS

We first like to note that the following lemma helps to quantify how much the distortion on transition probability is related to the distortion on the visitation probability.

**Lemma 2.** *If* $\max_{s,a} \|P(\cdot|s,a) - P^\dagger(\cdot|s,a)\|_1 \leq \frac{(1-\gamma)^2}{\gamma} \epsilon_d$ *where* $\epsilon_d > 0$*, then the agent can guarantee* $\epsilon_d$*-perceived visitation probability.*

We begin with Lemma 3 to prove Lemma 2. Recall that $P^{\dagger,\pi}(s,a)$ is the $\epsilon_d$-perceived visitation probability if $\max_{(s,a)} |P^\pi(s,a) - P^{\pi,\dagger}(s,a)| < \epsilon_d$. This perception gap arises from factors such as transition probabilities, policy, and state space. In the following lemma, we show how the perception gap in transition probability accumulates into the visitation probability. Before, we define $\epsilon_p$-perceived transition probability if $\max_{(s,a)} \|P(\cdot|s,a) - P^\dagger(\cdot|s,a)\|_1 < \epsilon_p$ holds. We denote $\mathbb{P}_t^\pi(s,a)$ as the probability of visiting $(s,a)$ at time $t$ with policy $\pi$.

**Lemma 3** (Bounding visitation probability of step $t$ when $\epsilon_p$-perceived transition holds)**.** *If for all* $(s,a)$ *holds* $\epsilon_p$*-perceived transition probability, then we have*

$$\max_\pi \left( \sum_{(s,a)\in\mathcal{S}\times\mathcal{A}} \left| \mathbb{P}_t^\pi(s,a) - \mathbb{P}_t^{\pi,\dagger}(s,a) \right| \right) \leq t\epsilon_p$$

*that holds for all* $t \in \mathbb{N}$

***Proof of Lemma*** *3.* Proof by induction. We use short notation for $P(s_t = s \mid s_{t-1} = s', a_{t-1} = a')$ as $P_t(s \mid s', a')$ and $P^\dagger(s_t = s \mid s_{t-1} = s', a_{t-1} = a')$ as $P_t^\dagger(s \mid s', a')$. By the definition of rational transition probability the statement holds at $t = 1$ for any policy $\pi$. Now, suppose the statement holds for $t - 1$ for any policy $\pi$. Then, we have

$$\sum_{(s,a)\in\mathcal{S}\times\mathcal{A}} \left| \mathbb{P}_t^\pi(s,a) - \mathbb{P}_t^{\pi,\dagger}(s,a) \right|$$

$$= \sum_{(s,a)\in\mathcal{S}\times\mathcal{A}} \left| \pi(a_t = a \mid s_t = s) \sum_{s',a'} \left( P_t(s \mid s', a') \mathbb{P}_{t-1}^\pi(s',a') \right) \right.$$

$$\left. - \pi(a_t = a \mid s_t = s) \sum_{s',a'} \left( P_t^\dagger(s \mid s', a') \mathbb{P}_{t-1}^{\pi,\dagger}(s',a') \right) \right|$$

$$\leq \sum_{(s,a)\in\mathcal{S}\times\mathcal{A}} \pi(a_t = a \mid s_h = s) \left| \sum_{s',a'} \left( P_t(s \mid s', a') \mathbb{P}_{t-1}^\pi(s',a') \right) - \sum_{s',a'} \left( P_t^\dagger(s \mid s', a') \mathbb{P}_{t-1}^{\pi,\dagger}(s',a') \right) \right|$$

$$= \sum_{s\in\mathcal{S}} \left| \sum_{s',a'} \left( P_t(s \mid s', a') \mathbb{P}_{t-1}^\pi(s',a') \right) - \sum_{s',a'} \left( P_t^\dagger(s \mid s', a') \mathbb{P}_{t-1}^{\pi,\dagger}(s',a') \right) \right|$$

$$= \sum_{s\in\mathcal{S}} \left| \sum_{s',a'} \left( P_t - P_t^\dagger \right) \mathbb{P}_{t-1}^\pi(s',a') + \sum_{s',a'} P_t^\dagger(s \mid s', a') \left( \mathbb{P}_{t-1}^\pi(s',a') - \mathbb{P}_{t-1}^{\pi,\dagger}(s',a') \right) \right|$$

$$\leq \sum_{s',a'} \left| \sum_{s\in\mathcal{S}} \left( P_t - P_t^\dagger \right) \mathbb{P}_{t-1}^\pi(s',a') \right| + \sum_{s',a'} \left| \sum_{s\in\mathcal{S}} P_t^\dagger(s \mid s', a') \left( \mathbb{P}_{t-1}^\pi(s',a') - \mathbb{P}_{t-1}^{\pi,\dagger}(s',a') \right) \right|$$

$$\leq \epsilon_p \sum_{s',a'} \mathbb{P}_{t-1}^\pi(s',a') + 1 \cdot (t-1)\epsilon_p$$

$$= \epsilon_p \cdot 1 + (t-1)\epsilon_p$$

$$\leq t\epsilon_p$$

The all of above inequalities hold for all $\pi$. Therefore, the statement holds for all $t \in \mathbb{N}$. □

Now, we prove the Lemma 2.

**Proof of Lemma 2.** Lemma 2 is almost a corollary that stems from Lemma 3. By the definition of visitation probability, we have

$$\sum_{(s,a)\in\mathcal{S}\times\mathcal{A}} \left|P^\pi(s,a) - P^{\pi,\dagger}(s,a)\right| = \sum_{(s,a)\in\mathcal{S}\times\mathcal{A}} \left|\sum_{t=0}^\infty \gamma^t \left(\mathbb{P}_t^\pi(s,a) - \mathbb{P}_t^{\dagger,\pi}(s,a)\right)\right|$$

$$\leq \sum_{(s,a)\in\mathcal{S}\times\mathcal{A}} \sum_{h=0}^\infty \gamma^t \left|\left(\mathbb{P}_t^\pi(s,a) - \mathbb{P}_t^{\dagger,\pi}(s,a)\right)\right|$$

$$= \sum_{t=0}^\infty \gamma^t \sum_{(s,a)\in\mathcal{S}\times\mathcal{A}} \left|\left(\mathbb{P}_t^\pi(s,a) - \mathbb{P}_t^{\dagger,\pi}(s,a)\right)\right|$$

$$\leq \sum_{h=0}^\infty \gamma^t t \frac{(1-\gamma)^2}{\gamma}\epsilon_p$$

Let $S = \sum_{t=0}^\infty \gamma^t t$, then $\gamma S = \sum_{t=0}^\infty \gamma^{t+1} t = \sum_{t=1}^\infty \gamma^t (t-1)$. Then by subtracting those two equations, we have $(1-\gamma)S = \sum_{t=1}^\infty \gamma^t = \frac{\gamma}{1-\gamma}$. Therefore we have $S = \frac{\gamma}{(1-\gamma)^2}$. Finally, we have the following inequality

$$\sum_{(s,a)\in\mathcal{S}\times\mathcal{A}} \left|P^\pi(s,a) - P^{\pi,\dagger}(s,a)\right| \leq \frac{\gamma}{(1-\gamma)^2}\cdot\frac{(1-\gamma)^2}{\gamma}\epsilon_p = \epsilon_p$$

$\square$

**Proof of Lemma 1.** First, note that we have assumed the image of the function $R$ is closed and dense as $[-R_{\max}, R_{\max}]$. Then, in the progress of projecting all $(s,a)$ into the reward, we define the probability of reward as $\mathbb{P}(\mathcal{R} = r) = \sum_{\forall(s,a)\in\mathcal{S}\times\mathcal{A}} d^\pi(s,a)\mathbf{1}[R(s,a) = r]$. we use short notation for $\mathbb{P}(\mathcal{R} = r)$ as $\mathbb{P}_{\mathcal{R}}$. Now, since $d^\pi(s,a)$ is the visitation probability of visiting $(s,a)$, then this could be converted to $\mathbb{P}(\mathcal{R} = r)$ by $d^\pi(\mathcal{R} = R^{-1}(s,a))$ where $R^{-1}$ is many to one function.

Now, since $\mathbb{R}$ is the many-to-one function, we can define independent block the $\mathcal{S}, \mathcal{A}$ as the set $Z(r) := \{(s,a) \in \mathcal{S} \times \mathcal{A} | R(s,a) = r\}$. Note that if $r_1 \neq r_2$, then $Z(r_1) \cap Z(r_2) = 0$. Then, if $h$ satisfies the set $Z$ to in be permutation-invariant. Namely, if $R(s_1, a) = R(s_2, a)$, then $R(h(s_1)) = R(h(s_2), a)$ holds then there exists a one-to-one mapping function $h : [-R_{\max}, R_{\max}] \to [-R_{\max}, R_{\max}]$ such that

$$R(s,a) = h(R(h(s), a))$$

holds. The proof can be divided into two folds. The existence of such a function and its one-to-one mapping function exists. We first prove the existence of such function $h$. This is because for any state and action $s, a$, suppose its reward value is $r$. Then suppose $g(s) = s'$. Then since image of function $R$ is closed and dense, there exists $r' \in [-R_{\max}, R_{\max}]$ such that $R(s', a) = r'$ holds. Then, one can say the function $r = h(r')$ exists. Now, we prove the one-to-one mapping property. suppose for two state and action pair $(s_1, a_1)$ and $(s_2, a_2)$ and let $s_1' = h(s_1')$ and $s_2' = h(s_2')$. Now, suppose $R(s_1', a) \neq R(s_2', a)$ holds. Then, due to the property of $h$, then it should also satisfy $R(s_1, a) \neq R(s_2, a)$. Therefore, this concludes that $h$ is the one-to-one mapping, and the following holds

$$d^\pi(R(g(s), a) = r) = d^\pi(h(R(g(s), a)) = h(r))$$
$$= d^\pi(R(s, a) = h(r))$$
$$= \mathbb{P}(\mathcal{R} = h(r))$$

holds. we denote $\mathbb{P}\left(\mathcal{R} = h(r)\right)$ as $\mathbb{P}_{h(\mathcal{R})}$. Then, let's define two different functions $h^+$ and $h^-$ such that we want to claim that

$$w^-\left(\int_{-R_{max}}^r d\mathbb{P}_{\mathcal{R}}\right) = \int_{-R_{max}}^r d\mathbb{P}_{h^-(\mathcal{R})}, \quad \text{and} \quad w^+\left(\int_{-R_{max}}^r d\mathbb{P}_{\mathcal{R}}\right) = \int_{-R_{max}}^r d\mathbb{P}_{h^+(\mathcal{R})} \tag{12}$$

holds for any $w^-, w^+$. Since the proof for either is similar, we prove the case for the existence of $h^-$ under $w^-$ distortion.

Now, recall that for $0 < x < b$, $w^-(x) < x$ holds and for $b < x < 1$, $w^-(x) > x$ holds and $w^-(x)$ is monotically increasing function. Define $r_b \in [-R_{\max}, 0]$ such that $b := \int_{-R_{\max}}^{r_b} d\mathbb{P}_{\mathcal{R}}$ holds, and for notation simplicity we deonte $F^-(r) = \int_{-R_{\max}}^{r_b} d\mathbb{P}_{\mathcal{R}}$. Then, one can say $-R_{\max} < r < r_b$, $w(F(r)) < F(r)$ holds and. Then we can always find a unique ratio $0 < \gamma(r) < 1$ that depends on $r$ such that $w^-(F(r)) = \int_{-R_{\max}}^{\gamma(r)r} d\mathbb{P}_r$ holds where

$$\gamma(r) = \frac{w^-(F(r))}{r}.$$

This leads to set $h(r) = \gamma(r)r = w^-(F(r))$ that satisfies (12) and also one-to-one mapping. In the same manner, we can also identify $h(r) = \gamma(r)r = w^-(F(r))$ where $r_b < r < 0$ holds for $\gamma(r) > 1$. Then, this completes that the function $h : r \to w^-(F(r))$ satisfies a one-to-one function and Equation (12). This completes the proof. $\qquad\square$ $\qquad\square$

***Proof of Theorem 1.*** By the definition of optimal policy and the value function definition at the time $T = 1$, we have the optimal policy at time $0$ as follows.

$$\pi^\star = \arg\max_\pi V_0(s)$$
$$= \arg\max_{a \in \mathcal{A}} Q_0(s, a)$$
$$= \arg\max_{a \in \mathcal{A}} R(s, a)$$
$$\pi^{\star,\dagger} = \arg\max_{a \in \mathcal{A}} V_0^\dagger(s)$$
$$= \arg\max_{a \in \mathcal{A}} Q_0^\dagger(s, a)$$
$$= \arg\max_{a \in \mathcal{A}} u(R(s, a))$$

for any fixed $s \in \mathcal{S}$, let's assume $a^*$ is the argument that maximizes the $R(s, a)$. Since $u$ is the non-decreasing convex function, $a^*$ is still the same argument that maximizes the $u(R(s, a))$. Therefore, $\pi^\star = \pi^{\star,\dagger}$ holds. $\qquad\square$ $\qquad\square$

***Proof of Theorem 2.*** We prove by backward induction. First by theorem 1, $\pi_T^\star = \pi_T^{\star,\dagger}$ holds. Now suppose that $\pi_{t'+1}^\star = \pi_{t'+1}^{\star,\dagger}$ holds for all $t' = t+1, \cdots, T$. Now, we prove the statement holds for $t$. To prove $\pi_t^\star = \pi_t^{\star,\dagger}$, it is sufficient to show if $Q_t^{\pi^*}(s, a) \geq Q_t^{\pi^*}(s, a')$, then $Q_t^{\dagger,\pi^*}(s, a) \geq Q_t^{\dagger,\pi^*}(s, a')$ also holds for any actions $a, a' \in \mathcal{A}$. First, the gap $Q_t^{\pi^*}(s, a) - Q_t^{\pi^*}(s, a)$ could be expressed as

$$Q_t^\pi(s, a) - Q_t^\pi(s, a) = R_t(s, a) - R_t(s, a') + \left\{\left(P(s_1|s, a) - P(s_2|s, a')\right)\left(V_{t+1}^{\pi^*}(s_1) - V_{t+1}^{\pi^*}(s_2)\right)\right\}$$
$$= \left(P(s_1|s, a) - P(s_2|s, a')\right)\left(V_{t+1}^{\pi^*}(s_1) - V_{t+1}^{\pi^*}(s_2)\right)$$

and $Q_t^{\dagger,\pi^\star}(s,a) - Q_t^{\dagger,\pi^\star}(s,a)$ as

$$Q_t^{\dagger,\pi^\star}(s,a) - Q_t^{\dagger,\pi^\star}(s,a) = R_t^\dagger(s,a) - R_t^\dagger(s,a') + \left\{\left(P^\dagger(s_1|s,a) - P^\dagger(s_2|s,a')\right)\left(V_{t+1}^{\pi^\star}(s_1) - V_{t+1}^{\pi^\star}(s_2)\right)\right\}$$

$$= \left(P^\dagger(s_1|s,a) - P^\dagger(s_2|s,a')\right)\left(V_{t+1}^{\dagger,\pi^\star}(s_1) - V_{t+1}^{\dagger,\pi^\star}(s_2)\right)$$

$$= \left(w(P^\dagger(s_1|s,a)) - w(P^\dagger(s_2|s,a'))\right)\left(V_{t+1}^{\dagger,\pi^\star}(s_1) - V_{t+1}^{\dagger,\pi^\star}(s_2)\right)$$

the reward during $t \in [1, T-1]$ is zero by our problem formulation assumption in section **??**. Now, without loss of generality, we assume $V_{t+1}^{\pi^\star}(s_1) > V_{t+1}^{\pi^\star}(s_2)$. Then, due to our assumption that $\pi_{t'}^\star = \pi_{t'}^{\star,\dagger}$ holds for $t' = t+1, \cdots, T$, we also have $V_{t+1}^{\dagger,\pi^\star}(s_1) > V_{t+1}^{\dagger,\pi^\star}(s_2)$. Also, noticing that weight function $w$ is also increasing function, then $P(s_1|s,a) > P(s_2|s,a)$ also guarantees $w(P(s_1|s,a)) > w(P(s_2|s,a))$ holds. Therefore, we can claim if $Q_t^\pi(s,a) - Q_t^\pi(s,a) > 0$ holds, then $Q_t^{\dagger,\pi^\star}(s,a) - Q_t^{\dagger,\pi^\star}(s,a) > 0$ also holds. Then, this leads to claim that $\arg\max Q_t^\pi(s,a) = \arg\max Q_t^{\dagger\pi}(s,a)$, which implies $\pi_t^\star = \pi_t^{\star,\dagger}$. This completes the proof. $\qquad\square$ $\qquad\square$

***Proof of Theorem 3***. Assume that Theorem 3 does not hold. Given $T = 2$, we have $V_2^\pi(s) = \max_{a\in\mathcal{A}} R_2(s,a) = R_2(s)$ for each state $s$. At time $t = 1$, assume $R_2(s_1) \le R_2(s_2) \le R_2(s_3)$. The condition $Q_1^{\dagger,\pi}(s,a_1) \ge Q_1^{\dagger,\pi}(s,a_2)$ is then expressed as:

$$w\left(P\left(s_1 \mid s, a_1\right)\right) r_2\left(s_1\right) + \left(w\left(P\left(s_2 \mid s, a_1\right) + P\left(s_1 \mid s, a_1\right)\right) - w\left(P\left(s_1 \mid s, a_1\right)\right)\right) R_2\left(s_2\right)$$
$$+ \left(1 - w\left(P\left(s_2 \mid s, a_1\right) + P\left(s_1 \mid s, a_1\right)\right)\right) R_3\left(s_3\right)$$
$$\ge w\left(P\left(s_1 \mid s, a_2\right)\right) R_2\left(s_1\right) + \left(w\left(P\left(s_2 \mid s, a_2\right) + P\left(s_1 \mid s, a_2\right)\right) - w\left(P\left(s_1 \mid s, a_2\right)\right)\right) R_2\left(s_2\right)$$
$$+ \left(1 - w\left(P\left(s_2 \mid s, a_2\right) + P\left(s_1 \mid s, a_2\right)\right)\right) R_3\left(s_3\right)$$

which simplifies to:

$$\left(w\left(P\left(s_1 \mid s, a_1\right)\right) - w\left(P\left(s_1 \mid s, a_2\right)\right)\right)\left(R_2\left(s_1\right) - R_3\left(s_3\right)\right)$$
$$+ \left(\left(w\left(P\left(s_2 \mid s, a_1\right) + P\left(s_1 \mid s, a_1\right)\right) - w\left(P\left(s_1 \mid s, a_1\right)\right)\right)\right.$$
$$\left. - \left(w\left(P\left(s_2 \mid s, a_2\right) + P\left(s_1 \mid s, a_2\right)\right) - w\left(P\left(s_1 \mid s, a_2\right)\right)\right)\right)\left(R_2\left(s_2\right) - R_3\left(s_3\right)\right) \ge 0$$

For the non-distorted case, the analogous expression is:

$$\left(P\left(s_1 \mid s, a_1\right) - P\left(s_1 \mid s, a_2\right)\right)\left(R_2\left(s_1\right) - R_3\left(s_3\right)\right)$$
$$+ \left(P\left(s_2 \mid s, a_1\right) - P\left(s_2 \mid s, a_2\right)\right)\left(R_2\left(s_2\right) - R_3\left(s_3\right)\right) \ge 0$$

For arbitrary reward functions, $R_2$, the equality of the two cases under any weighting function $w$ leads to:

$$\frac{w\left(P\left(s_1 \mid s, a_1\right)\right) - w\left(P\left(s_1 \mid s, a_2\right)\right)}{w\left(P\left(s_2 \mid s, a_1\right) + P\left(s_1 \mid s, a_1\right)\right) - w\left(P\left(s_1 \mid s, a_1\right)\right) - \left(w\left(P\left(s_2 \mid s, a_2\right) + P\left(s_1 \mid s, a_2\right)\right) - w\left(P\left(s_1 \mid s, a_2\right)\right)\right)}$$
$$= \frac{P\left(s_1 \mid s, a_1\right) - P\left(s_1 \mid s, a_2\right)}{P\left(s_2 \mid s, a_1\right) - P\left(s_2 \mid s, a_2\right)}$$

where $w(p) = p$ is the only solution, contradicting the distortion required by Definition 2. $\qquad\square$

***Proof of Theorem 4***. The proof of Theorem 4 is divided into three-fold.

### 1. Proof of asymptotic convergence

We first prove Equation (3) of Theorem 4 in this part 1, then we prove Equation (4) of Theorem 4 in part 3 of this proof. Note that the empirical distribution function $\widehat{F}_n(r)$ generate Stielgies measure which takes mass $\frac{1}{t}$ each of the sample points on $U^+(R_i)$.

or equivalently, show that

$$\lim_{n \to +\infty} \sum_{i=1}^{n-1} u^+(R_{[i]})(w^+(\frac{n-i+1}{n}) - w^+(\frac{n-i}{n})) \xrightarrow{n \to \infty} \int_0^{+\infty} w^+(P(U > t))dt, \text{w.p. } 1 \qquad (13)$$

where $n$ denotes the number of positive reward among $|\mathcal{S}||\mathcal{A}|$. Let $\xi_{\frac{i}{n}}^+$ and $\xi_{\frac{i}{n}}^-$ denote the $\frac{i}{n}$th quantile of $u^+(X)$ and $u^-(X)$, respectively.

For the convergence proof, we first concentrate on finding the following probability,

$$P\left(\left|\sum_{i=1}^{n-1} u^+(R_{[i]}) \cdot \left(w^+\left(\left(\frac{n-i}{n}\right) - w^+\left(\frac{n-i-1}{n}\right)\right) - \sum_{i=1}^{n-1} \xi_{\frac{i}{n}}^+ \cdot \left(w^+\left(\frac{n-i}{n}\right) - w^+\left(\frac{n-i-1}{n}\right)\right)\right)\right| > \epsilon\right),$$
$$(14)$$

for any given $\epsilon > 0$. It is easy to check that

$$P(\left|\sum_{i=1}^{n-1} u^+(R_{[i]}) \cdot (w^+(\frac{n-i}{n}) - w^+(\frac{n-i-1}{n})) - \sum_{i=1}^{n-1} \xi_{\frac{i}{n}}^+ \cdot (w^+(\frac{n-i}{n}) - w^+(\frac{n-i-1}{n}))\right| > \epsilon)$$

$$\leq P(\bigcup_{i=1}^{n-1} \left\{\left|u^+(R_{[i]}) \cdot (w^+(\frac{n-i}{n}) - w^+(\frac{n-i-1}{n})) - \xi_{\frac{i}{n}}^+ \cdot (w^+(\frac{n-i}{n}) - w^+(\frac{n-i-1}{n}))\right| > \frac{\epsilon}{n}\right\})$$

$$\leq \sum_{i=1}^{n-1} P(\left|u^+(R_{[i]}) \cdot (w^+(\frac{n-i}{n}) - w^+(\frac{n-i-1}{n})) - \xi_{\frac{i}{n}}^+ \cdot (w_{(}^+\frac{n-i}{n}) - w_{(}^+\frac{n-i-1}{n}))\right| > \frac{\epsilon}{n}) \qquad (15)$$

$$= \sum_{i=1}^{n-1} P(\left|(u^+(R_{[i]}) - \xi_{\frac{i}{n}}^+) \cdot (w^+(\frac{n-i}{n}) - w^+(\frac{n-i-1}{n}))\right| > \frac{\epsilon}{n})$$

$$\leq \sum_{i=1}^{n-1} P(\left|(u^+(R_{[i]}) - \xi_{\frac{i}{n}}^+) \cdot (\frac{1}{n})^\alpha\right| > \frac{\epsilon}{n})$$

$$= \sum_{i=1}^{n-1} P(\left|(u^+(R_{[i]}) - \xi_{\frac{i}{n}}^+)\right| > \frac{\epsilon}{\cdot n^{1-\alpha}}). \qquad (16)$$

The right-hand side of Inequality (16) could be expressed as follows.

$$P\left(\left|u^+(R_{[i]}) - \xi_{\frac{i}{n}}^+\right| > \frac{\epsilon}{n^{(1-\alpha)}}\right)$$
$$= P\left(u^+(R_{[i]}) - \xi_{\frac{i}{n}}^+ > \frac{\epsilon}{n^{(1-\alpha)}}\right) + P\left(u^+(R_{[i]}) - \xi_{\frac{i}{n}}^+ < -\frac{\epsilon}{n^{(1-\alpha)}}\right).$$

We focus on the term $P\left(u^+(R_{[i]}) - \xi^+_{\frac{i}{n}} > \frac{\epsilon}{n^{1-\alpha}}\right)$. Now, let us define an event $A_t = I_{(u^+(X_t) > \xi^+_{\frac{i}{n}} + \frac{\epsilon}{n^{(1-\alpha)}})}$ where $t = 1, \ldots, n$. Since the Cumulative distribution is non-decrasing function, we have the following,

$$P\left(u^+(R_{[i]}) - \xi^+_{\frac{i}{n}} > \frac{\epsilon}{1-\alpha}\right) = P\left(\sum_{t=1}^n A_t > n \cdot (1 - \frac{i}{n^{(1-\alpha)}})\right)$$

$$= P\left(\sum_{t=1}^n A_t - n \cdot [1 - F^+(\xi^+_{\frac{i}{n}} + \frac{\epsilon}{n^{(1-\alpha)}})] > n \cdot [F^+(\xi^+_{\frac{i}{n}} + \frac{\epsilon}{n^{(1-\alpha)}}) - \frac{i}{n}]\right).$$

Using the fact that $\mathbb{E}A_t = 1 - F^+(\xi^+_{\frac{i}{n}} + \frac{\epsilon}{n^{(1-\alpha)}})$ in conjunction with Hoeffding's inequality, we obtain

$$P(\sum_{i=1}^n A_t - n \cdot [1 - F^+(\xi^+_{\frac{i}{n}} + \frac{\epsilon}{n^{(1-\alpha)}})] > n \cdot [F^+(\xi^+_{\frac{i}{n}} + \frac{\epsilon}{n^{(1-\alpha)}}) - \frac{i}{n}]) < e^{-2n \cdot \delta'_t}, \tag{17}$$

where $\delta'_i = F^+(\xi^+_{\frac{i}{n}} + \frac{\epsilon}{n^{(1-\alpha)}}) - \frac{i}{n}$. Since $F^+(x)$ is Lipschitz, we have that $\delta'_i \leq L_{F^+} \cdot (\frac{\epsilon}{1-\alpha})$. Hence, we obtain

$$P(u^+(R_{[i]}) - \xi^+_{\frac{i}{n}} > \frac{\epsilon}{1-\alpha}) < e^{-2n \cdot L_{F^+} \frac{\epsilon}{1-\alpha}} = e^{-2n^\alpha \cdot L^+ \epsilon} \tag{18}$$

In a similar fashion, one can show that

$$P(u^+(R_{[i]}) - \xi^+_{\frac{i}{n}} < -\frac{\epsilon}{1-\alpha}) \leq e^{-2n^\alpha \cdot L_{F^+} \epsilon} \tag{19}$$

Combining (18) and (19), we obtain

$$P(\left|u^+(R_{[i]}) - \xi^+_{\frac{i}{n}}\right| > \frac{\epsilon}{1-\alpha}) \leq 2 \cdot e^{-2n^\alpha \cdot L_{F^+} \epsilon}, \ \forall i \in \mathbb{N} \cap (0,1)$$

Plugging the above in (16), we obtain

$$P(\left|\sum_{i=1}^{n-1} u^+(R_{[i]}) \cdot (w^+(\frac{n-i}{n}) - w^+(\frac{n-i-1}{n})) - \sum_{i=1}^{n-1} \xi^+_{\frac{i}{n}} \cdot (w^+(\frac{n-i}{n}) - w^+(\frac{n-i-1}{n}))\right| > \epsilon)$$
$$\leq 2n \cdot e^{-2n^\alpha \cdot L_{F^+}}. \tag{20}$$

Notice that $\sum_{n=1}^{+\infty} 2n \cdot e^{-2n^\alpha \cdot L_{F^+} \epsilon} < \infty$ since the sequence $2n \cdot e^{-2n^\alpha \cdot L_{F^+}}$ will decrease more rapidly than the sequence $\frac{1}{n^k}, \forall k > 1$.

By applying the Borel Cantelli lemma, we have that $\forall \epsilon > 0$

$$P(\left|\sum_{i=1}^{n-1} u^+(R_{[i]}) \cdot (w^+(\frac{n-i}{n}) - w^+(\frac{n-i-1}{n})) - \sum_{i=1}^{n-1} \xi^+_{\frac{i}{n}} \cdot (w^+(\frac{n-i}{n}) - w^+(\frac{n-i-1}{n}))\right| > \epsilon) = 0,$$

which implies

$$\sum_{i=1}^{n-1} u^+(R_{[i]}) \cdot (w^+(\frac{n-i}{n}) - w^+(\frac{n-i-1}{n})) - \sum_{i=1}^{n-1} \xi^+_{\frac{i}{n}} \cdot (w^+(\frac{n-i}{n}) - w^+(\frac{n-i-1}{n})) \xrightarrow{n \to +\infty} 0 \ \text{w.p 1},$$

which proves (13).

Also, the remaining part, conducting the proof of convergence of $w^-$ and $u^-$, i.e.

$$\lim_{n \to +\infty} \sum_{i=1}^{n-1} u^-(R_{[i]})(w^-(\frac{n-i+1}{n}) - w^-(\frac{n-i}{n})) \xrightarrow{n \to \infty} \int_0^{+\infty} w^-(P(U > t))dt, \text{w.p. } 1 \tag{21}$$

also follows simliar manner. we omit the proof for this.

**2. Proof of value function lower bound**

By the definition, we have the following

$$\begin{aligned}|V_{\mathcal{M}}(s_0) - V_{\mathcal{M}^\dagger}(s_0)| &= \left|\int_{-\infty}^{0} w_\star^-(\mathbb{P}_r(u_\star^-(\mathcal{R} > r)))dr - \int_{\infty}^{0} w^-(\mathbb{P}_r(u^-(\mathcal{R} > r)))dr\right| \\ &= \left|\int_{-\infty}^{0} w_\star^-(\mathbb{P}_r(u_\star^-(\mathcal{R} > r)))dr - \int_{\infty}^{0} w_\star^-(\mathbb{P}_r(u^-(\mathcal{R} > r)))dr \right. \\ &\quad \left. -\left(\int_{\infty}^{0} w^-(\mathbb{P}_r(u^-(\mathcal{R} > r)))dr - \int_{\infty}^{0} w_\star^-(\mathbb{P}_r(u^-(\mathcal{R} > r)))dr\right)\right| \\ &\geq \underbrace{\left|\int_{-\infty}^{0} w_\star^-(\mathbb{P}_r(u_\star^-(\mathcal{R} > r)))dr - \int_{-\infty}^{0} w_\star^-(\mathbb{P}_r(u^-(\mathcal{R} > r)))dr\right|}_{\text{Term (I)}} \\ &\quad -\underbrace{\left|\int_{-\infty}^{0} w_\star^-(\mathbb{P}_r(u_\star^-(\mathcal{R} > r)))dr - \int_{\infty}^{0} w_\star^-(\mathbb{P}_r(u^-(\mathcal{R} > r)))dr\right|}_{\text{Term (II)}}\end{aligned}$$

We first under bound the term (I). For notation simplicity, we let $g(r) = \mathbb{P}_r(u^-(\mathcal{R} > r)))$ and $g_\star(r) = \mathbb{P}_r(u_\star^-(\mathcal{R} > r)))$. Then we have the following

$$\text{Term (I)} = \left|\int_{-R_{\max}}^{0} w_\star^-(g_\star(r)) - w_\star^-(g(r))\right|$$

Now, since $w_\star^-(x)$ is monotonically increasing in $x \in [0, a]$ and monotonically decreasing in $x \in [a, 1]$, we could say for any $x, y \in [0, 1], x \neq y$ that

$$\frac{w_\star^-(x) - w_\star^-(y)}{x - y} = (w_\star^-)'(z) \geq \min_{z \in [0,1]} (w_\star^-)'(z) = \min\left\{(w_\star^-)'(0), (w_\star^-)'(1)\right\},$$

where $z \in (x, y)$. The first equality holds due to the mean value theorem. Therfore it holds that

$$\begin{aligned}\text{Term (I)} &= \left|\int_{-R_{\max}}^{0} w_\star^-(g_\star(r)) - w_\star^-(g(r))\right| \\ &\geq \left|\int_{-R_{\max}}^{0} \min\left\{(w_\star^-)'(0), (w_\star^-)'(1)\right\}(g_\star(r) - g(r))\right| \\ &= \min\left\{(w_\star^-)'(0), (w_\star^-)'(1)\right\}\left|\int_{-R_{\max}}^{0} (g_\star(r) - g(r))\right|\end{aligned}$$

Now, recall the definition of $g_\star(r)$ and $g(r)$, then we have the following

$$\left|\int_{-R_{\max}}^{0} (g_\star(r) - g(r))\, dr\right| = |\mathbb{E}_{\mathcal{R} \sim \mathbb{P}_\pi}[u_\star^-(\mathcal{R}) - u^-(\mathcal{R})]|$$

Now, let us denote the intersection of $u^-(R)$ and $y = R + C_{bs}$ as $R = -R_{bs}$. We can say if the blackswan happens, then its reward is bounded between $[-R_{\max}, -R_{bs}]$. Then we have the following,

$$\left| \int_{-R_{\max}}^{0} (g_\star(r) - g(r)) \right| = \left| \mathbb{E}_{\mathcal{R} \sim \mathbb{P}_\pi} \left[ u_\star^-(\mathcal{R}) - u^-(\mathcal{R}) \right] \right|$$

$$= \left| \mathbb{E}_{\mathcal{R} \sim \mathbb{P}_\pi} \left[ \mathbf{1} \left[ \mathcal{R} < -R_{bs} \right] (u_\star^-(\mathcal{R}) - u^-(\mathcal{R})) \right] \right.$$

$$\left. - \mathbb{E}_{\mathcal{R} \sim \mathbb{P}_\pi} \left[ \mathbf{1} \left[ \mathcal{R} \geq -R_{bs} \right] (-u_\star^-(\mathcal{R}) + u^-(\mathcal{R})) \right] \right|$$

$$\geq \underbrace{\left| \mathbb{E}_{\mathcal{R} \sim \mathbb{P}_\pi} \left[ \mathbf{1} \left[ \mathcal{R} < -R_{bs} \right] (u_\star^-(\mathcal{R}) - u^-(\mathcal{R})) \right] \right|}_{\text{Term I-1}}$$

$$- \underbrace{\left| \mathbb{E}_{\mathcal{R} \sim \mathbb{P}_\pi} \left[ \mathbf{1} \left[ \mathcal{R} \geq -R_{bs} \right] (-u_\star^-(\mathcal{R}) + u^-(\mathcal{R})) \right] \right|}_{\text{Term I-2}}$$

$$\geq \left| \mathbb{E}_{\mathcal{R} \sim \mathbb{P}_\pi} \left[ \mathbf{1} \left[ \mathcal{R} < -R_{bs} \right] (u_\star^-(\mathcal{R}) - u^-(\mathcal{R})) \right] \right|$$

To lower bound the Term I-1, let's denote the minimum reachability of blackswan events as $\epsilon_{bs}^{\min} \neq 0$. Then we have

$$\text{Term I-1} \geq \frac{R_{\max} - R_{bs}}{R_{\max}} \epsilon_{bs}^{\min} \min_{R \in [-R_{\max}, -R_{bs}]} |u^-(R) - u_\star^-(R)|$$

$$\geq \frac{R_{\max} - R_{bs}}{R_{\max}} \epsilon_{bs}^{\min} |u^-(-R_{bs}) - u_\star^-(-R_{bs})| \tag{22}$$

$$\text{Term I-2} \leq \frac{R_{bs}}{R_{\max}} \epsilon_{bs} \max_{R \in [-R_{bs}, 0]} |u^-(R) - u_\star^-(R)|$$

$$\leq \frac{R_{bs}}{R_{\max}} \epsilon_{bs} |u^-(-R_{bs}) - u_\star^-(-R_{bs})| \tag{23}$$

Therefore, we have the following equation,

$$\text{Term I} \geq \frac{(R_{\max} - R_{bs}) \epsilon_{bs}^{\min} - R_{bs} \epsilon_{bs}}{R_{\max}} |u^-(-R_{bs}) - u_\star^-(-R_{bs})|$$

Also, since the function $u_\star^-(r)$ is convex, and $u_\star^-(-R_{\max}) < -R_{\max} + C_{bs}$ holds. Therefore, we could say $u_\star^-(r) < \frac{R_{\max} - C_{bs}}{R_{\max}} r$. This leads us to come up with $u_\star^-(-R_b s) < \frac{R_{\max} - C_{bs}}{R_{\max}} (-R_{bs})$. Therefore, we have a gap lowerbound as

$$|u^-(-R_{bs}) - u_\star^-(-R_{bs})| \geq (R_{\max} - C_{bs}) \frac{R_{bs}}{R_{\max}} - (R_{bs} - C_{bs})$$

$$= \frac{(R_{\max} - R_{bs}) C_{bs}}{R_{\max}}$$

The above inequality could be minimized as

$$\text{Term I} \geq \frac{(R_{\max} - R_{bs}) \, \epsilon_{bs}^{\min} - R_{bs} \epsilon_{bs}}{R_{\max}} \left( \frac{(R_{\max} - R_{bs}) C_{bs}}{R_{\max}} \right)$$

$$= \frac{\left( (R_{\max} - R_{bs}) \, \epsilon_{bs}^{\min} - R_{bs} \epsilon_{bs} \right) (R_{\max} - R_{bs}) C_{bs}}{R_{\max}^2}$$

Now, let's upper bound Term 2. Before, recall that the definition of $g(r) = \mathbb{P}_r(u^-(\mathcal{R}) > r))$ and note that by the definition of black swans, we have $u^-(\mathcal{R}) > \mathcal{R} + C_{bs}$ holds for $R \in [-R_{\max}, -R_{bs})$. Therefore, we can say for all $r \in [-R_{\max}, -R_{bs}), g(r) = 1$ holds. Therefore, for all $r \in [-R_{\max}, -R_{bs}]$, we have $w_\star^-(g(r)) - w^-(g(r)) = w_\star^-(1) - w^-(1) = 1 - 1 = 0$

$$\left| \int_{-R_{\max}}^0 w_\star^-(g(r)) - w^-(g(r)) dr \right| = \left| \int_{-R_{\max} + C_{bs}}^0 w_\star^-(g(r)) - w^-(g(r)) dr \right|$$

$$= \left| \int_{-R_{\max} + C_{bs}}^0 w^-(g(r)) - w_\star^-(g(r)) dr \right|$$

$$\leq \left| \int_{-R_{\max} + C_{bs}}^0 L^- g(r) - g(r) dr \right|$$

$$= (L^- - 1) \left| \int_{-R_{\max} + C_{bs}}^0 g(r) dr \right|$$

$$\leq (L^- - 1) \cdot \frac{R_{\max} - C_{bs}}{2 R_{\max}} \epsilon_{bs}$$

$$= (L^- - 1) \left| \int_{-R_{\max} + C_{bs}}^0 1 - \mathbb{P}_r(U^-(\mathcal{R}) < r) dr \right|$$

$$= (L^- - 1) \left| \int_{-R_{\max} + C_{bs}}^0 1 - \mathbb{P}_r(U^-(\mathcal{R}) < r) dr \right|$$

$$= (L^- - 1) \left| \left( (R_{\max} - C_{bs}) - \int_{-R_{\max} + C_{bs}}^0 \mathbb{P}_r(u^-(\mathcal{R}) < r) dr \right) \right|$$

$$= (L^- - 1) \left| \left( (R_{\max} - C_{bs}) - \mathbb{E}_{\mathcal{R} \sim \mathbb{P}_r} \left[ u^-(\mathcal{R}) \mathbf{1} [-R_{\max} + C_{bs} < \mathcal{R} < 0] \right] \right) \right| \tag{24}$$

Note that if $-R_{\max} + C_{bs} < -R_{bs}$, then

$$\mathbf{1} [-R_{\max} + C_{bs} < \mathcal{R} < 0] \cdot \mathbb{E}_{\mathcal{R} \sim \mathbb{P}_r} \left[ u^-(\mathcal{R}) \right] \geq \left( \frac{R_{\max} - C_{bs} - R_{bs}}{2 R_{\max}} \epsilon_{bs}^{\min} + \frac{R_{bs}}{2 R_{\max}} \epsilon_{bs} \right) u^-(-R_{\max} + C_{bs}) \tag{25}$$

and if $-R_{\max} + C_{bs} < -R_{bs}$, then

$$\mathbf{1} [-R_{\max} + C_{bs} < \mathcal{R} < 0] \cdot \mathbb{E}_{\mathcal{R} \sim \mathbb{P}_r} \left[ u^-(\mathcal{R}) \right] \geq \left( \frac{R_{\max} - C_{bs}}{2 R_{\max}} \epsilon_{bs} \right) u^-(-R_{\max} + C_{bs}) \tag{26}$$

Therefore, combining the Equations (24), (25), (26), we conclude that

$$\text{Term II} \leq C \cdot \frac{\left( (R_{\max} - R_{bs}) \, \epsilon_{bs}^{\min} - R_{bs} \epsilon_{bs} \right) (R_{\max} - R_{bs}) C_{bs}}{R_{\max}^2}$$

where $C \in [0, 1]$ is a constant. This completes the proof.

### 3. Value function upper bound

For the proof of Equation (4) of Theorem 4, we utilized the following Lemma 4 which provides a concentration inequality on the distance between empirical distribution and true distribution.

Since $u^+(\mathcal{R})$ is bounded above by $u^+(R_{\max})$ and $w^+(p)$ is Lipschitz with constant $L^+(= (w^+)'(a))$, we have the following inequality,

$$\left| \int_0^\infty w^+(P(u^+(X)) > x) dx - \int_0^\infty w^+(1 - \hat{F}_t^+(x)) dx \right|$$

$$= \left| \int_0^{u^+(R_{\max})} w^+(P(u^+(X)) > x) dx - \int_0^{u^+(R_{\max})} w^+(1 - \hat{F}_t^+(x)) dx \right|$$

$$\leq \left| \int_0^{u^+(R_{\max})} L^+ \cdot |P(u^+(X) < x) - \hat{F}_t^+(x)| dx \right|$$

$$\leq L^+ u^+(R_{\max}) \sup_{x \in \mathbb{R}} \left| P(u^+(X) < x) - \hat{F}_t^+(x) \right|.$$

Now, plugging in the DKW inequality, we obtain

$$P\left( \left| \int_0^\infty w^+(P(u^+(X)) > x) dx - \int_0^\infty w^+(1 - \hat{F}_t^+(x)) dx \right| > \epsilon/2 \right)$$

$$\leq P\left( L^+ u^+(R_{\max}) \sup_{x \in \mathbb{R}} \left| (P(u^+(X) < x) - \hat{F}_t^+(x) \right| > \epsilon/2 \right) \leq 2 e^{-t \frac{\epsilon^2}{2(L^+ u^+(R_{\max}))^2}}. \tag{27}$$

Along similar manner, we have

$$P\left( \left| \int_0^\infty w^-(P(u^-(X)) > x) dx - \int_0^\infty w^-(1 - \hat{F}_t^-(x)) dx \right| > \epsilon/2 \right) \leq 2 e^{-t \frac{\epsilon^2}{2(L^- u^-(-R_{\max}))^2}}. \tag{28}$$

Combining (27) and (28), we obtain

$$P(|V_{\overline{\mathcal{M}}^\dagger} - V_{\mathcal{M}^\dagger}| > \epsilon) \leq P\left( \left| \int_0^\infty w^+(P(u^+(X)) > x) dx - \int_0^\infty w^+(1 - \hat{F}_t^+(x)) dx \right| > \epsilon/2 \right)$$

$$+ P\left( \left| \int_0^\infty w^-(P(u^-(X)) > x) dx - \int_0^\infty w^-(1 - \hat{F}_t^-(x)) dx \right| > \epsilon/2 \right)$$

$$\leq 4 e^{-t \frac{\epsilon^2}{2c^2}}.$$

where $c = \max\{|L^+ u^+(R_{\max})|, |L^- u^-(-R_{\max})|\}$

□ □

***Proof of Theorem 5.*** For a given optimal policy $\pi_\star$, define the normalized occupancy measure as $d_{\pi_\star} = (1 - \gamma) \sum_{t=0}^\infty \gamma^t \mathbb{P}_\pi((s_t, a_t) = (s, a))$. Note that $d_{\pi_\star}$ represents the stationary distribution. Additionally, given the assumption that the reward function $R : \mathcal{S} \times \mathcal{A} \to \mathbb{R}$ is a bijection, it follows that the distribution $d_{\pi_\star}(R^{-1}(s, a))$ and $\mathbb{P}_r$ are identical. This indicates that the occurrence of black swan events can be entirely characterized by the reward values, rather than the specific state-action pairs.

Now, we define the event $E_{bs} := \{\mathcal{R} \in [-R_{\max}, -R_{bs}]\}$ where $\mathcal{R} \sim \mathbb{P}_r$. The probability of event $E_{bs}$ happens is bounded as follows

$$
\begin{aligned}
\mathbb{P}(E_{bs}) &= F(-R_{bs}) - F(-R_{\max}) \\
&= F(-R_{bs}) \\
&\in \left( \left( \frac{R_{\max} - R_{bs}}{2R_{\max}} \right) \epsilon_{bs}^{\min}, \left( \frac{R_{\max} - R_{bs}}{2R_{\max}} \right) \epsilon_{bs}^{\max} \right) \\
&:= [p_{bs}^{\min}, p_{bs}^{\max}]
\end{aligned}
$$

Note that we have assumed the $0 < \mathbb{P}_r(r = R(s,a)) < \epsilon_{bs}$ and its minimum reachable probability as $\epsilon_{bs}^{\min}$ for all reward. now, for given trajectory, the reward instance is given as $(r_1, r_2, ...r_h, ...)$ where $r_h \sim \mathbb{P}_r$, the probability that the agent first visit the black swan event at step $h$ would be defined as

$$
\begin{aligned}
\mathbb{P}(r_1, \cdots, r_{h-1} \notin E_{bs}, r_h \in E_{bs}) &= (1 - \mathbb{P}(E_{bs}))^{h-1} \mathbb{P}(E_{bs}) \\
&\leq (1 - p_{\min})^{h-1} p_{\max}
\end{aligned}
$$

Therefore, its probability is bounded as follows,

$$
(1 - p_{\max})^{h-1} p_{\min} \leq \mathbb{P}(r_1, \cdots, r_{h-1} \notin E_{bs}, r_h \in E_{bs}) \leq (1 - p_{\min})^{h-1} p_{\max}
$$

Now, to ensure that the blackswan probability to be lower bounded than $\delta$, we need the following conditions,

$$
\begin{aligned}
\delta &\leq (1 - p_{\max})^{h-1} p_{\min} \\
\log \delta &\leq (h-1) \log(1 - p_{\max}) + \log p_{\min}
\end{aligned}
$$

Therefore, we have

$$
h \geq \log(\delta/p_{\min}) / \log(1 - p_{max}) + 1.
$$

Therefore, we can conclude that if $h = \Omega(\log(\delta/p_{\min}) / \log(1 - p_{max}))$, then the agent's probability to meet the black swan is at least $\delta$.

$\square$ $\qquad\qquad\qquad\qquad\qquad\qquad\qquad\qquad\qquad\qquad\qquad\qquad\qquad$ $\square$

## H   HELPFUL LEMMAS

**Lemma 4.** *(Dvoretzky-Kiefer-Wolfowitz (DKW) inequality)*
*Let $\hat{F}_n(u) = \frac{1}{n} \sum_{i=1}^{n} 1_{((u(X_i)) \leq u)}$ denote the empirical distribution of a r.v. U, with $u(X_1), \ldots, u(X_n)$ being sampled from the r.v $u(X)$. The, for any $n$ and $\epsilon > 0$, we have*

$$
P(\sup_{x \in \mathbb{R}} |\hat{F}_n(x) - F(x)| > \epsilon) \leq 2e^{-2n\epsilon^2}.
$$

