# OpenReview forum: "A Black Swan Hypothesis: The Role of Human Irrationality in AI Safety"
_ICLR.cc/2025/Conference — ICLR 2025 Poster_

### Official Review · Reviewer_234c · 2024-10-22

**Soundness:** 3
**Presentation:** 2
**Contribution:** 2
**Rating:** 5
**Confidence:** 3

**Summary:**

This paper proposes that rare and high-risk events, namely black swans, can originate from misperceptions of an event’s reward and likelihood even in static environments. The paper formalizes this concept in the context of MDP-based decision tasks using the machineary of cumulative prospect theory, where the misperceptions of rewards and transition functions in MDPs are characterized by distortion functions, resulting in a gap between the ground truth MDP and the MDP perceived by humans. The paper then theoretically analyzes the impact of black swans on the value function and the hitting time of black swan events.

**Strengths:**

The main proposal of the paper, namely the black swan events can emerge in stationary environments, is interesting and makes sense to me. I also found the formalization using distortion functions on transition probabilities and rewards natural and captures the intuition of human irrationality.

**Weaknesses:**

In my opinion, the most important weakness of the paper is that apart from the main claim (black swans in unchanging environments), it is not clear to me what the main takeaways of the paper are. The authors wrote in the abstract "We hope these definitions can pave the way for the development of algorithms to prevent such events by rationally correcting limitations in perception", but it is not clear to me how the definitions and theoretical results presented in the paper can really benefit this: under the formalization of the paper, black swans in stationary environments essentially stem from the distorted transition and reward functions, which naturally create a gap between the ground truth MDP and the human MDP. The main theoretical result of the paper (Theorem 4) focuses on proving this gap in the context of value function estimation, yet such a result is quite straightforward and the techniques used in proving it (bounding the gap in value estimation given the gap in transition and reward functions) is also somewhat common in the RL literature. Perhaps more importantly, I feel that such a result (and also the result in Theorem 5) is not really useful to inspire algorithm design since it does not tell us _how_ to correct the misperceptions in the human MDP. Other results in the paper are also quite natural to me and seem not really tied to the specific black swan problem considered by the paper (see Questions for details). While the authors have extensively discussed in the appendix that existing solutions in safe RL may also fall short, only _defining_ such a problem is not enough to me for acceptance.

I also have some concerns about the overall presentation of the paper. For example, the formal definition of s-black swan is deferred to Section 6 but heavily referred to in earlier sections, and I think it may be better to move this definition to earlier sections in the paper. I also do not understand the role of Section 3 beyond it re-emphasizes the main claim of the paper in Remark 1.

**Questions:**

- Is the proof of Lemma 1 in the appendix correct? The lemma in the main text is stated for visitation probabilities, while the proof in the appendix seems to only deal with rewards. The authors could clarify if there is a connection between the reward-based proof and the visitation probability statement that is not immediately apparent.

- Theorem 3 seems a quite general (and somewhat trivial) result to me: of course if we consider a sufficiently general case, the difference in MDP transition probabilities would result in different optimal policies. What is the concrete relation between this result and the s-black swan defined by the paper?

- What is the role of Proposition 1？And isn't it contradictory to Algorithm 1 (Algorithm 1 defines an s-black swan for arbitrary $t$, while Proposition 1 considers $t$ in specific time intervals)?

- How the definitions and theoretical results in the paper may be used to aid algorithm design?

- Could the hitting time analysis in Theorem 5 be used to inform when to trigger perception updates?

---

> ### Author Response · Authors · 2024-11-19
> **Thanks for the comments!**
>
> Dear Reviewer 234c,
>
> First, we would like to thank the reviewer for their fruitful and thoughtful comments. We are impressed by the depth of the feedback, which reflects a thorough scrutiny of our draft and demonstrates the significant time and effort invested. Before addressing reviewer's concerns, we kindly encourage the reviewer to first read the **global comment on the main message**, followed by our responses that address specific weaknesses and questions.
>
> We are happy to provide further clarification on any remaining concerns or questions from the reviewer.
>
>
>  **Q1.Proof of Lemma1**
>
> Thanks for pointing out the gap between Lemma and the proof. ***Yes, the proof of Lemma 1 is correct***, and we apologize for any confusion. We understand that, at first glance, there may appear to be a significant gap between the statement and the proof of Lemma 1, which could be challenging for readers to bridge. For the analysis, we ***transform the visitation probability of a specific state-action pair $(s, a)$ into the visitation probability of the corresponding reward, as described on line 1131 to 1133 of the appendix***. We believe not clarifying this point creates a large gap in understanding the proof and the Lemma1. This transformation allows us to perform further analysis in the space of $\mathbb{R}$, rather than in the more complex space of $\mathcal{S} \times \mathcal{A}$. This shift simplifies the problem and facilitates a more tractable analysis.
>
> **Q2.Trivialness of Theorem 3**
>
>  We appreciate your observation regarding this matter. The main reason why Theorem $3$ is introduced is to align the statements of Theorems $1$ and $2$ with the ***completeness of Section $4$***. We understand the reviewer's point -- we are happy to rewrite Theorem $3$ as a Remark to avoid over-emphasizing the contribution of this paper.
>
> **Q3.Role of Proposition 1**
>
>  Also, thanks for brining this up. We would like to first clarify that the role of Section $3$ is to classify black swans in stationary environments from black swans in non-stationary environments. Since a stationary environment is a subset of non-stationary environments, we first state this classification through Algorithm $1$. Then, Proposition $1$ states that for any black swan event, we can always find a time interval that classifies any black swans in S-blackswan (which can also be thought of as an observation interval). The ***main role of Proposition $1$ is to establish the possibility of applying our analysis (interpretation) to any black swans***. To be more specific, suppose that $(s, a, t)$ is a black swan for $t \in [10, 20]$. EBy Algorithm $1$, even though $(s, a)$ is a black swan in a non-stationary environment, this paper also demonstrates that it is possible to classify $(s, a)$ as an S-black swan during the time interval $[10, 20]$, enabling further interpretation and methods to handle $(s, a)$ during $[10, 20]$. This is also well illustrated in Example $2$.
>
> **Q4.Aiding Algorihtm design**
>
> Thanks for asking this question. *We believe that this is a crucial point*. We also believe that the global comments provided above partially address the reviewer's concerns. To be more specific, the lower bound of S-black swan events depends on three factors: *greater distortion* in reward perception (i.e., larger $C_{bs}$), *a larger feasible set* for S-black swan (i.e., larger $R_{\max} - R_{bs}$), and a *higher minimum probability* of S-black swan occurrence (i.e., larger $\epsilon^{\min}_{bs}$). This framework highlights the importance of designing future algorithms that aim to correct human perception by reducing reward distortion. Additionally, these algorithms should also focus on decreasing the feasible set size and the minimum probability of occurrence. ***This is still an open direction but to the best of our knowledge, such considerations can inform how much the agent should explore in a safe manner that takes misperception in a consideration.That is, our analysis enables guideing the design of an safe exploration strategy that is sensitive to whether the probability or the size of the feasible set is being enlarged.*** This connection between exploration strategies and the underlying factors provides a pathway for advancing the design of algorithms to handle S-black swan events.
>
> **Q5.Hitting time analysis**
>
> Also, thanks for pointing out the implication of the hitting time analysis. This is well-stated in lines 429 to 432, which explain how frequently the correction algorithm should be updated in proportion to the magnitude of the perception gap and the minimum frequency of black swan events.

---

> > ### Comment · Reviewer_234c · 2024-11-21
> > **Response of the reviewer**
> >
> > Thank you for your detailed response, which addressed some of my concerns. I agree with you that the central message of the paper---that black swans can emerge due to human misperception, is an interesting and potentially significant contribution. I would also like to make it clear that my initial review is not meant to criticize this main message, but to share some of my concerns on (1) the formalization of the main message and (2) the implications of such a formalization.
> >
> > In particular, I now realize that my initial feeling mostly came from the fact that most of MDP analysis in the paper models black-swans as _distortions on rewards and transition functions_ (e.g., Defs. 4 and 5). While this formalization is indeed natural, I think the current exposition of the paper does not really makes it stand out from the conventional analysis on MDPs, since "changing the reward distribution and/or transition probabilities can change the optimal policy/value estimation" is a well-known result in RL theory. I would also think that even the conventional (non-static) black swans can also be modeled as "distortions" on rewards or transition probs, despite that such distortions are caused by the non-stationary MDP dynamics itself. But what is the essential difference between the analysis of the conventional black swans and the s-black swans then?
> >
> > To address this concern, I think the authors could include some concrete examples (e.g., in an application scenario) to underline the importance of formalizing the s-black swans, and discuss how such a formalization can indeed make a difference compared with the conventional formalization of black swans.
> >
> > Some follow-up questions that I still have:
> >
> > - Q1. I still do not understand why transforming the visitation probability of a specific state-action pair into the visitation probability of the corresponding reward is correct---as you have mentioned in the proof, different state-action pair may be mapped to the same reward, how can we discriminate between them then?
> >
> > - Q3. I also do not understand why _any_ black swans can also be modeled by your framework---does this mean that there is not an essential difference between black swans and the introduced s-black swans? But then the position of the paper would be questionable.

---

> ### Author Response · Authors · 2024-11-21
> **Thanks for the review**
>
> Dear Reviewer 234c,
>
> Thanks for getting back to us. Actually, we are delighted to note that the points you raised present an excellent opportunity to enhance the paper's contribution, beyond simply conveying its central message (as discussed earlier).
>
> To address your concerns, we have provided detailed responses organized into the following four key points: [Weakness 1, 2 and Question 1,2]
>
> We believe this discussion has brought us to a shared understanding, allowing us to respond to your queries in a more direct and precise manner. Should you have any additional questions or concerns, please do not hesitate to reach out -- we would be more than happy to continue the discussion!
>
> ### ***[Weakness 1] S-black swan is just an emergence from a non-zero model error from true $P$ and $R$, so the theoretical analysis is trivial, making it indistinguishable from prior RL theory literature focusing on PAC bound analysis via model estimation.***
>
> $\textcolor{red}{\text{This is really great question!}}$.
>
> To address this, we believe it is helpful to reframe the question into a more fundamental inquiry as follows:
>
>  * ***[Question]: How does CPT-distortion on the model (P,R) provide an additional interpretation of the suboptimality gap?"***
>
> Our answer is
>
>  * ***[Answer]: "CPT-induced distortions exhibit an order invariance property, meaning that they preserve the rank order of $P$ and $R$, while still allowing for the possibility of finding suboptimal policies (as demonstrated in Theorem 3)."***
>
> Let us elaborate further:
>
> Theorems 1, 2, and 3 in Section 4 collectively clarify the distinction. The main message of Theorems 1 and 2 is that in environments with low complexity, the agent selects the same optimal policy, as the distortion functions $w$ and $v$ preserve the order of $R(s, a)$ and $P^\pi(s, a)$ for all $(s, a)$. Then, the key difference from prior PAC-bound analyses lies in the following:
>
>  * *[Detailed answer]: Regardless of the degree of distortion in $R$ and $P^\pi$ (i.e., irrespective of how large the model-error bound $\epsilon_r,\epsilon_p$ is, in terms of RL theoretical literature), in low-complexity environments, the agent will still select the true optimal policy due to the "order-invariant" property of $w$ and $v$ (as outlined in Theorems 1 and 2). However, this "order invariance" breaks as the complexity of the environment increases, leading to suboptimal policy selection, which is highlighted in Theorem 3.*
>
>  ### **(Additional answer1) Additional explanation for reviewer's understanding on [Weakness 1]**
>
> We hope the explanation above is helpful. However, if further clarification is needed, we would like to provide additional support by bringing in Theorem 4 to address your concerns more comprehensively. To frame this discussion more specifically, another key question might be:
>
> "How does the lower bound of the value gap provide a different perspective beyond the typical interpretation involving $\epsilon_r$ and $\epsilon_p$ as model errors?"
>
> In Theorem 4, while the lower bound does include the distortion of reward $C_{bs}$—a factor that aligns with previous RL theory literature utilizing $\epsilon_r$—it also introduces two additional factors:
>
>  * A larger feasible set for S-black swan events (i.e., $R_{max} - R_{bs}$ increases).
>  * A higher minimum probability of S-black swan occurrence (i.e., $\epsilon^{min}_{bs}$ is larger).
>
> Even though both of these factors diminish as $\epsilon_p$ converges to zero, we believe they represent new, distinct elements of the lower bound. These factors capture important characteristics of the problem that are not fully encapsulated by $\epsilon_p$. By considering these elements explicitly, the lower bound offers a richer perspective on the dynamics of S-black swan events, beyond merely interpreting them as aspects of model error.

---

> ### Author Response · Authors · 2024-11-21
>
> ### **(Additional answer2) Experiment to support our answer for [Weakness 1]**
>
> To further support the reviewer’s insightful question regarding the impact of **CPT-characterized distortion** on the suboptimality gap, we would like to provide simulation results addressing this point. If we agree that the critical question is:
>
> **"How does CPT-characterized distortion provide an *additional meaning* to the suboptimal gap?"**
>
> then a natural follow-up question arises:
>
> **"[Question] Does CPT-characterized distortion offer any *benefit* to the agent in finding the optimal policy?"**
>
> To explore this, we conducted additional experiments. Consider an 8x8 Grid World where the agent always starts at $(0, 0)$ and can reach one of six different goal states located at $(4, 4)$, $(5, 5)$, $(6, 6)$, $(7, 7)$, $(8, 8)$, and $(9, 9)$. Each goal has a distinct reward from the set $[0.1, 1, 10, 100, 1000, 10000]$, and the agent incurs a step penalty of $-0.1$ for each move. Naturally, the optimal policy under real-world conditions would be to reach the goal at $(9, 9)$, which provides the highest reward.
>
> Interestingly, our simulation results reveal that there exists a **specific pair of reward distortion and probability distortion** (as defined in Definitions 1 and 2) that enables the agent to discover a better policy than what was optimized under real-world conditions.
>
> ### Experimental Setup
>
> - **Distorted Worlds:** We consider six distorted environments, all of which share the same reward distortion pattern that adheres to Definitions 1 and 2.
> - **Distortion Parameter ($\gamma$):** $\gamma$ represents the distortion rate of transition probabilities, where smaller values of $\gamma$ indicate larger distortions.
> - **Algorithm:** All experiments were performed using the Q-learning algorithm.
> - **Outcome Metric:** The probabilities represent the visitation frequency of each goal state.
>
> ### Results
>
> The results demonstrate that CPT-distorted worlds can enhance policy selection by leveraging the interaction between reward and probability distortions, ultimately leading to improved agent behavior in some cases.
> ### Table1
> | visitation probability of Goal State      | (4,4)   | (5,5)   | (6,6)   | (7,7)   | (8,8)   | (9,9) = Optimal goal   |
> |-----------------|---------|---------|---------|---------|---------|---------|
> | **(1) Real world (no distortion)**                   | 0.87929 | 0.09817 | 0.01621 | 0.00414 | 0.00168 | **0.00051** |
> | **(2) Distorted world (only reward distortion)** | 0.93762 | 0.05177 | 0.00728 | 0.00207 | 0.00090 | **0.00036** |
> | **(3) Distorted world (gamma=0.9 with reward distortion)**            | 0.92069 | 0.06538 | 0.00986 | 0.00264 | 0.00108 | **0.00035** |
> | **(4) Distorted world (gamma=0.8  with reward distortion)**            | 0.90552 | 0.07670 | 0.01244 | 0.00342 | 0.00147 | **0.00044** |
> | **(5) Distorted world (gamma=0.7  with reward distortion)**            | 0.87729 | 0.09861 | 0.01674 | 0.00482 | 0.00185 | **0.00070** |
> | **(6) Distorted world (gamma=0.6  with reward distortion)**            | 0.83943 | 0.12043 | 0.02770 | 0.00843 | 0.00309 | **0.00090** |
> | **(7) Distorted world (gamma=0.5  with reward distortion)**            | 0.81693 | 0.13363 | 0.03255 | 0.01048 | 0.00466 | **0.00175** |
>
> It is easy to observe that for a fixed reward distortion, as the distortion of transition probability increases (i.e., smaller $\gamma$), the distorted world shows a higher probability of visiting the state $(9,9)$ than the real-world scenario. For example, in the real-world, the visitation probability is $0.00051$, while in the distorted world, it surpasses this value when $\gamma < 0.7$ (e.g., at $\gamma = 0.7$, the probability is $0.00070$).
>
> So, why does this happen? Intuitively, this occurs because **reward distortion alone** causes the agent to perceive the step reward of $-0.1$ as a significantly lower value, such as $-5$. This perception incentivizes the agent to prioritize shorter trajectories to avoid accumulating large penalties from step rewards, potentially leading to suboptimal goals (that is going to among (4,4) to (8,8).
>
> However, **distorting probabilities** introduces exploration, encouraging the agent to consider a broader range of trajectories. This additional exploration allows the agent to better evaluate distant goals, often leading to improved policies.  Therefore, our answer to the above question is
>
> **[Answer]: CPT-distortion provide an intrinsic motivation (or help shape a beneficial intrinsic reward) for the agent to find the optimal policy that that of real-world**

---

> ### Author Response · Authors · 2024-11-21
>
> (continue) We have checked that CPT distribution can effectively serve this purpose. Specifically, we find that there exists a **reward bonus term** based on count-based exploration, expressed as:
>
> $$r(s,a) + \frac{\beta}{\sqrt{n(s, a)}}, $$
>
> where $n(s, a)$ represents the visitation count of state-action pair $(s, a)$.
>
> ### Table 2
> | visitation probability of Goal State          | (4,4)   | (5,5)   | (6,6)   | (7,7)   | (8,8)   | (9,9)   |
> |-------------------|---------|---------|---------|---------|---------|---------|
> | **(1) Real world (no distortion)**                   | 0.87929 | 0.09817 | 0.01621 | 0.00414 | 0.00168 | 0.00051 |
> | **(8) Real world (Reward Bonus - beta=0.5)**                   | 0.86549 | 0.10520 | 0.02041 | 0.00587 | 0.00235 | 0.00068 |
> | **(9) Real world (Reward Bonus  - beta=1.0)**                   | 0.85266 | 0.11078 | 0.02397 | 0.00817 | 0.00340 | 0.00103 |
> | **(10) Real world (Reward Bonus  - beta=1.5)**                   | 0.83812 | 0.11758 | 0.02782 | **0.01000** | **0.00479** | **0.00170** |
> | **(11) Real world (Reward Bonus  - beta=2.0)**                   | 0.82782 | 0.12016 | 0.03137 | 0.01253 | 0.00587 | 0.00215 |
> | **(12) Real world (Reward Bonus  - beta=3.0)**                   | 0.80600 | 0.12934 | 0.03732 | 0.01605 | 0.00831 | 0.00298 |
> | **(7) Distorted world (gamma=0.5)**            | 0.81693 | 0.13363 | 0.03255 | **0.01048** | **0.00466** | **0.00175** |
>
> From above table, please check that the bonus term when $\beta = 1.5$ in the real-world aligns with the best performance observed in the distorted world (gamma=0.5).
>
> We hope this explanation provides a clear intuition behind the mechanism of CPT distortion and its potential to shape intrinsic motivation!
>
> Should you have any further questions or require clarification, we would be delighted to discuss this further :)

---

> ### Author Response · Authors · 2024-11-21
>
> ## **[Weakness 2]. Distortion Can Also Be Regarded as Non-Stationarity**
>
> Thank you for raising this question -- this is indeed an excellent point! We agree that the distortion of an MDP between time $t$ and $t+1$ is mathematically similar to the distinction between a Ground-MDP and a Human-MDP. However, while the two formulations are mathematically analogous, they provide different perspectives on the design of RL algorithms.
>
> In non-stationary environments, the majority of RL algorithms focus on how to adapt quickly to previously unseen environments. On the other hand, viewing the gap as a form of **static misperception** offers an alternative approach. It suggests finding a "static" distortion function that can eventually debias the collected data prevents a blackswan. This is also similar to think about finding the "stationary source (function v and w)" that generates the non-stationary environments.
>
> The following example might not be perfect, but our view can also be applied to identify optimal factors that reduce the non-stationarity of the output we aim to predict.
>
> ### Example: Predicting a Patient's Blood Sugar Level
>
> Suppose we aim to predict a patient’s blood sugar level, denoted as $A$, as a function of factors $W$, $X$, $Y$, and $Z$—such as the level of carbohydrate intake or the severity of the patient’s illness. We design a prediction function:
>
> $$ A = f(W, X, Y, Z), $$
>
> and train $f$ on a given dataset of input-output pairs, $((w, x, y, z), a)$.
>
> However, due to human misperception about the relevant factors, we might overlook an important variable, $V$, which significantly impacts blood sugar spikes (i.e., black swan events). Specifically, humans might misperceive the joint probability $p(V, A)$ as being zero, and incorrectly assign the reward $r(V, A)$ as $r(V, A) + 100$, perceiving it as a benign event. As a result, $V$ is excluded when designing the function $f$.  Despite our best efforts, failing to include $V$ in the model design introduces errors or non-stationarity in the predicted blood sugar levels, particularly for extreme cases.
>
> This example illustrates how missing critical factors due to human misperception can lead to inaccurate predictions. Furthermore, addressing such misperceptions could potentially reduce the variance (or non-stationarity) of the output $A$, improving model reliability and performance.
>
>
> ## **[Question 1]: Transforming the Visitation Probability of a Specific State-Action Pair into the Visitation Probability of the Corresponding Reward**
>
> Thank you for this thoughtful question! We agree that establishing a one-to-one mapping between state-action visitation probabilities and reward values is not highly practical setting. However, we believe this is a trivial technical issue. Recall that "Humans" manually design the reward of certain events, which are not inherently given by the environment as true values. It is entirely open and flexible for a **Human reward designer** to craft rewards corresponding to different state-action pairs. For example, they could:
>
> 1. Add Gaussian noise to rewards in a sparse reward setting.
> 2. Design the reward function as a continuous function at the initial stage.
>
> This flexibility allows for practical implementation in various environments that enables the one-to-one mapping.
>
> ## **[Question 2]: What Is the Meaning of Any Black Swan?**
>
> We apologize for the earlier confusion—let us clarify this point more specifically.
>
> What we meant by saying "any black swan can be regarded as an S-Black Swan" depends on how the **time interval** is chosen. Recall that any non-stationary environment can be conceptualized as a **piecewise stationary environment**. In this framework, a general non-stationary environment that changes at every time step $t$ can be viewed as a time interval of 1 (i.e., a stationary environment for each time step).
>
> Thus, our statement means that for a fixed time interval where certain segments of the non-stationary environment are stationary, a black swan can be regarded as an S-Black Swan within that interval.
>
> #### Practicality of This Setting
> This setting is practical because, in many cases, non-stationarity evolves slowly over time. For example:
>
> - **User Preferences:** Preferences for products or movies do not change every second or every day, making them piecewise non-stationary reward functions.
> - **Insulin Reaction for Diabetes Patients:** The effect of insulin depends on the condition of a patient's body, but the condition does not fluctuate every second or every day, making it a piecewise non-stationary probability function.
>
> This concept is well-illustrated in **Case 2 of Example 2**.
>
> We hope this detailed explanation addresses your concerns. Please feel free to reach out with additional questions or for further clarification - we are more than happy to discuss!

---

> > ### Comment · Reviewer_234c · 2024-11-23
> > **Reviewer's follow-up response**
> >
> > Thank you for the very detailed response. At this point, I think you have addressed many of my initial concerns. In particular, I found the clarification of Theorem 4 and the order-preserving property of CPT-distortion helpful. I also like the discussion on the difference between static and dynamic distortions and believe that incorporating the above clarifications into the paper will benefit its future readership.
> >
> > Some further comments on the authors' follow-up response:
> >
> > - On the experiment: I think the experiment you conducted is indeed interesting, but I am not sure about whether it is on theme: anyway your main argument is that CPT-distortion is bad for finding the optimal policy (such that it can be used to model "black swans"), so providing an example showing that some distortions are actually beneficial is somewhat strange (although also making sense) to me.
> >
> > - On the example of predicting blood sugar level: I think this example is interesting and indeed reflects the s-black swan notion that you attempted to formalize. Yet, it would be better if you could take a step further and show how your formalization can help in addressing this problem---perhaps in a future version of the paper (e.g., how to find a "debiasing function" to correct the distortions). I know that this may still require some additional efforts but do believe that this could be a great addition to your work.
> >
> > In general, I appreciate the efforts of the authors in the response and decided to raise my rating from 3 to 5.

---

> ### Author Response · Authors · 2024-11-23
> **Follow-up response for Reviewer**
>
> Dear Reviewer 234c,
>
> Thank you for your active engagement in these discussions! Your thoughtful feedback has been immensely valuable in helping us clarify and refine the presentation of our paper’s contributions. We have prepared follow-up responses to address potential misunderstandings and further address your concerns. As always, we are more than happy to provide additional clarifications if needed.
>
> ---
>
> ## **[Point 1: does CPT-distortion really helps the agent to find optimal policy?]**
>
> It seems there may have been some misunderstandings, and we would like to clarify an important point:
>
> **CPT-distortion can actually help the agent discover the optimal policy.**
>
> Our experimental results support this claim, along with additional experiments suggesting that the underlying reason is that CPT-distortion may provide an intrinsic motivation (or reward) [3,4,5] for the agent to find the optimal policy. This phenomenon, while counterintuitive, has been noted as an interesting and impactful insight in prior work [1, 2]. For example, paper [1] explicitly states in its abstract:
>
> > *"Irrationality fundamentally helps rather than hinders reward inference, but it needs to be correctly accounted for."*
>
> ### Experimental Clarification
>
> To provide greater clarity, we would like to revisit and further elaborate on the results of our experiments mentioned earlier.  For better clarity, we have numbered the entries in **Table 1** and **Table 2** as (1) through (12).
>
> #### **Key Insights from Table 1:**
> - Results in **Table 1** show that as the level of distortion increases (moving from (2) → (3) → ... → (7)), the probability of visiting the optimal state $(9,9)$ also increases, surpassing the real-world case (denoted as (1)).
> - **Why does this happen?**
>
> #### **Key Insights from Table 2:**
> - Results in **Table 2** empirically show that increasing the portion of an exploration bonus (a common method for designing intrinsic rewards/motivation [4]) also increases the visitation probability of $(9,9)$ (moving from (8) → (9) → ... → (12)).
> - Notably, **case (10) matches with case (7)**, providing strong evidence that CPT-distortion can effectively serve as an intrinsic motivation mechanism for the agent to find the optimal policy.
>
> We hope this explanation helps clarify the misunderstanding. Please let us know if you have any further questions or need additional clarification!
>
> ## **[Point2: regarding with algorithm design]**
>
> Thank you for your support and thoughtful engagement with the sugar-level example. We are pleased to hear that it helps clarify the distinction between non-stationarity and model distortion, addressing the reviewer’s concerns. Indeed, a natural extension of this work would be to explore future algorithm designs that aim to debias the distortion function. While we have not included algorithm design in this draft, we hope the current manuscript effectively conveys the **main message** we have emphasized and demonstrates its contribution to the field.  Please let us know if further clarification would be helpful - we are always happy to discuss!
>
>
>
> ## Reference
> [1] Chan, Lawrence, Andrew Critch, and Anca Dragan. "Human irrationality: both bad and good for reward inference." arXiv preprint arXiv:2111.06956 (2021).
> [2] Kwon, Minae, Erdem Biyik, Aditi Talati, Karan Bhasin, Dylan P. Losey, and Dorsa Sadigh. "When humans aren't optimal: Robots that collaborate with risk-aware humans." In Proceedings of the 2020 ACM/IEEE international conference on human-robot interaction, pp. 43-52. 2020.
> [3] Chentanez, Nuttapong, Andrew Barto, and Satinder Singh. "Intrinsically motivated reinforcement learning." Advances in neural information processing systems 17 (2004).
> [4] Strehl, Alexander L., and Michael L. Littman. "An analysis of model-based interval estimation for Markov decision processes." Journal of Computer and System Sciences 74.8 (2008): 1309-1331.
> [5] Bellemare, M., Srinivasan, S., Ostrovski, G., Schaul, T., Saxton, D., & Munos, R. (2016). Unifying count-based exploration and intrinsic motivation. Advances in neural information processing systems, 29.

---

> ### Author Response · Authors · 2024-12-02
>
> Dear Reviewer 234c,
>
> We sincerely thank you for your active participation, which has greatly contributed to making this draft more insightful, enhancing our ability to convey our message to the readers -- especially great questions and follow-up as experiments section. As we approach the end of the rebuttal phase, we wanted to kindly ask if you have any additional concerns or feedback that we can address. Once again, thank you for your valuable engagement!
>
> Best,
> Authors

---

### Official Review · Reviewer_WW6X · 2024-11-01

**Soundness:** 3
**Presentation:** 3
**Contribution:** 2
**Rating:** 6
**Confidence:** 2

**Summary:**

This paper is a theory paper that challenges the view that black swan events only originate from changing (non-stationary) environments. Instead, the paper focuses on defining S-Black Swan events, which occur in unchanging environments due to human misperception of events’ values and probabilities. The paper is focused on formalizing the definition of S-Black-Swan events, by starting from Hypothesis 1 in the introduction.

**Strengths:**

* The main arguments of the paper are well-structured and well-communicated. The flow of the paper is helpful to the reader in communicating both the preliminary materials as well as leading to the mathematical formulation of the S-Black-Swan definition. For a theory paper, which could have the tendency to overcomplicate results, it feels like the authors have made significant effort to make the paper readable and therefore potentially meaningful to those who might use it as a future reference.
* The definitions of the stationary ground MDP, the Human MDP, and the Human Estimation MDP form a clear picture of the potential Agent-Environment Framework that could lead to a S-Black-Swan event.
* The presentation showing that black-swan events can arise in stationary environments is interesting.

**Weaknesses:**

* One weakness of the paper is in its ability to build a strong link between the results of the paper and how it might affect the wider machine learning community. This weakness can be broken down into a combination of the following:
	*  The related works section is left to the end and reads a bit like a list of works at the intersection of expected utility theory and reinforcement learning. The reader gets to the end of the section and is then told that this literature does not cover black swan events, but this statement lacks enough motivation.
	* The contribution of the paper is highlighted as defining an S-Black Swan event, but this contribution does not appear to be motivated by issues the current RL algorithms struggle within the literature. Every so often the paper includes comments like “aimed at guiding the design of safer ML algorithms in the future”; “laying the groundwork for future algorithmic design”; and “traditional risk criteria in RL are insufficient for managing the unique risks associated with black swan events”. While these comments might be the motivation of the paper, a more concrete motivation could be showing (or referencing) a specific RL (or ML) scenario which could benefit from this new definition and formalization of a S-Black-Swan event. This update seems like it would be important for an ICLR conference venue.

Typos:
* Line 108 $\mathbf{p}_c = …$ should be 3 prob choices.

Comment:
* Figures 1c and 1d could benefit from being moved further down in the paper.

**Questions:**

1. Are there any examples where a machine learning study/problem/algorithm would have benefited from the definition of a S-Black-Swan event?
2. The probability distortion function seems like it would be an easier function to measure in practice compared to the value distortion function. Each individual might legitimately have different reasons to value outcomes differently. Going with Example 1, a loss of -1000 might be significantly worse for a poorer person than a richer person. It looks like the theory still holds when $\epsilon_r = 0$, but $\epsilon_d > 0$, but could the authors comment on that?
3. If possible, could the authors provide additional context to the novelty of the work and how previous works have not considered an S-Black-Swan event, and attributed rare events to changing environments.

---

> ### Author Response · Authors · 2024-11-19
> **Thanks for the comments!**
>
> Dear Reviewer WW6X,
>
> We would like to sincerely thank the reviewer for requesting additional evidence and highlighting the novelty of our work. Before addressing the specific concerns, we kindly encourage the reviewer to first review the global comment on the **Main Message**, followed by our detailed responses to specific weaknesses and questions as below!

---

> ### Author Response · Authors · 2024-11-19
>
> **Q1. Real-world examples that can benefit from the definition of S-Black Swan events**
>
> We appreciate the reviewer’s concern regarding our hypothesis - specifically, whether interpreting black swan events through the lens of human misperception is meaningful and applicable in real-world scenarios. To provide additional clarity and support for this perspective, we offer further examples beyond the Lehman Brothers bankruptcy case mentioned in the introduction:
>
> * **[Case 1: Unexpected Drowning of NASA Astronauts Due to Overlooked Details]**
> Before launching rockets, NASA conducted tests on its space suits using high-altitude hot-air balloons. On May 4, 1961, Victor Prather and another pilot ascended to 113,720 feet to evaluate the suit's performance. While the test itself was successful, an unforeseen risk led to tragedy during the planned ocean landing. Prather opened his helmet faceplate to breathe fresh air, and as he slipped into the water while attaching to a rescue line, his now-exposed suit filled with water. Despite NASA’s rigorous planning and preparation, the risk of opening the faceplate - perceived as an extremely minor detail - was ***underestimated, resulting in catastrophic consequences***. This highlights how rigorously meticulous planning can still fail to account for overlooked events. [1,2]
>
> References:
> [1] Jan Herman, “Stratolab: The Navy’s High-Altitude Balloon Research,” lecture, Naval Medical Research Institute, Bethesda, MD, 1995, archive.org/details/StratolabTheNavysHighAltitudeBalloonResearch.
> [2] Douglas Brinkley, American Moonshot (New York: Harper, 2019), 237.
>
> * **[Case 2: Healthcare - Hypoglycemic Events in Diabetes Patients]**
> Diabetes patients typically experience a highly chronic condition, making their state relatively predictable a few hours into the future (stationary environment). However, ***rare hypoglycemic events***, characterized by a sudden and dangerous drop in blood sugar, pose significant risks. To address this, [3] developed an RL model capable of predicting changes in a patient's condition and providing optimized treatments. Furthermore, [4, 5] emphasize the critical importance of selecting appropriate signals as inputs, as ***human misperceptions about what constitutes important signals can lead to unexpected and suboptimal decisions***. As a result, traditional supervised learning methods may struggle to accurately predict rare events like hypoglycemic episodes when the input signals fail to align with the underlying dynamics.
>
> Reference:
>
> [3] Wang, G., Liu, X., Ying, Z. et al. Optimized glycemic control of type 2 diabetes with reinforcement learning: a proof-of-concept trial. Nat Med 29, 2633–2642 (2023). https://doi.org/10.1038/s41591-023-02552-9.
> [4] Panda, D., Ray, R., & Dash, S.R. (2020). Feature Selection: Role in Designing Smart Healthcare Models. Intelligent Systems Reference Library.
> [5] C. Ambhika, S. Gayathri, A. T. P, B. G. Sheena, N. M and S. S. R, "Enhancing Predictive Modeling in High Dimensional Data Using Hybrid Feature Selection," 2024 5th International Conference on Electronics and Sustainable Communication Systems (ICESC),
>
> **Q2. Different distortion with respect to different individuals**
>
> Thank you for raising this insightful question—it is an excellent point! We agree that individuals may exhibit varying distortions in value and perception. Addressing how black swan events arise due to the collective behavior in a multi-agent human setting, where each individual has a distinct distortion rate, is an intriguing avenue for future work. Such differences could potentially result in varying suboptimal policies, further highlighting the complexity of this phenomenon.
>
> **Q3. Additional context on the novelty and prior works that have not considered S-Black Swan events**
>
> * **Additional context on the novelty**:
>   Thank you for pointing out the importance of clarifying the novelty of this work. We believe that our global comment on the **Main Message** provides a precise explanation of the context and contributions of this study. However, if this is insufficient, please let us know - we would be happy to provide further elaboration.
>
> * **Prior works that have not considered S-Black Swan events**:
>   We believe that our response to the reviewer's previous question, **[Q1. Real-world examples that can benefit from the definition of S-Black Swan events]**, address this issue. Please feel free to let us know if additional clarification is needed, and we will be glad to expand further.
>
> **W1. How the results of the paper help the AI community**
>
> We kindly refer the reviewer to our response to **[Q4. Aiding Algorithm Design] from Reviewer 234c** and our global comment on the **Main Message** for insights on how our results contribute to future algorithm design and how our work can positively influence the AI community.

---

> ### Author Response · Authors · 2024-11-23
> **Further Clarifications on the Reviewer's Concerns**
>
> Dear Reviewer WW6X,
>
> We hope our response addresses the reviewer’s initial concerns! If Reviewer WW6X has any further points or questions, please feel free to share them with us. Your valuable initial feedback has helped us to:
>
>  * present empirical evidence (a gridworld multi-goal experiment) that supports our main message,
>  * provide an additional real-world example illustrating how a static misperception can lead to black swan events,
>  * clarify the novelty of our work in relation to the global comments summarized as the "main message," and
>  * acknowledge the potential extension of our work to multi-agent settings where agents exhibit differing distortion rates, which we can highlight as future work.
>
> We look forward to hearing your thoughts on whether these revisions have addressed your concerns. Please let us know if further clarification is needed, and we would be glad to continue the discussion.
>
> Best,
> Authors

---

> > ### Comment · Reviewer_WW6X · 2024-11-26
> > **Response**
> >
> > Thanks for your response. I am not sure that the provided "real-world examples" tackle the original main weakness that I put forward: "ability to build a strong link between the results of the paper and how it might affect the wider machine learning community". The NASA example and the Healthcare example do not link this work to RL.

---

> ### Author Response · Authors · 2024-11-26
>
> Dear Reviewer WW6X,
>
> Thank you for your thoughtful follow-up comment. We sincerely appreciate your understanding and we are writing further comments to address some points that may have been misunderstood.
>
> ### **[Q1] How to build a strong link between the results of the paper and how it might affect the wider machine learning community?**
>
> Thank you for raising this important question. We agree that establishing a strong connection between our findings and their implications for the wider machine learning community is crucial for deepening the contribution of our work. Our response to this question aligns closely with our answer to [W1] from Reviewer 234c, and we modify our previous comments to address this context:
>
> To address this question, we will answer how the model error induced by CPT-distortion can provide a new message in RL theory as follows :
>
>  * ***[Question]: How does CPT-distortion on the model (P, R) provide an additional interpretation of the suboptimality gap?"***
>
> Our answer is
>
>  * ***[Answer]: "CPT-induced distortions exhibit an order invariance property, meaning that they preserve the order of $P$ and $R$, while still allowing for the possibility of finding suboptimal policies (as demonstrated in Theorem 3)."***
>
> Let us elaborate further:
>
> Theorems 1, 2, and 3 in Section 4 collectively clarify the distinction. The main message of Theorems 1 and 2 is that in environments with low complexity, the agent selects the same optimal policy, as the distortion functions $w$ and $v$ preserve the order of $R(s, a)$ and $P^\pi(s, a)$ for all $(s, a)$. Then, the key difference from prior PAC-bound analyses lies in the following:
>
>  * *[Detailed answer]: Regardless of the degree of distortion in $R$ and $P^\pi$ (i.e., irrespective of how large the model-error bound $\epsilon_r,\epsilon_p$ is, in terms of RL theoretical literature), in low-complexity environments, the agent will still select the true optimal policy due to the "order-invariant" property of $w$ and $v$ (as outlined in Theorems 1 and 2). However, this "order invariance" breaks as the complexity of the environment increases, leading to suboptimal policy selection, which is highlighted in Theorem 3.*
>
> Furthermore, maybe a more interesting question is "How does the lower bound of the value gap provide a different perspective beyond the typical interpretation involving $\epsilon_r$ and $\epsilon_p$ as model errors?"
>
> In Theorem 4, while the lower bound does include the distortion of reward $C_{bs}$—a factor that aligns with previous RL theory literature utilizing $\epsilon_r$—it also introduces two additional factors:
>
>  * A larger feasible set for S-black swan events (i.e., $R_{max} - R_{bs}$ increases).
>  * A higher minimum probability of S-black swan occurrence (i.e., $\epsilon^{min}_{bs}$ is larger).
>
> Even though both of these factors diminish as $\epsilon_p$ converges to zero, we believe they represent new, distinct elements of the lower bound. These factors capture important characteristics of the problem that are not fully encapsulated by $\epsilon_p$. By considering these elements explicitly, the lower bound offers a richer perspective on the dynamics of S-black swan events, beyond merely interpreting them as aspects of model error.
>
> ### Additional experiment to support Q1
>
> To further support our answer tothe  reviewer's comments, we would like to provide simulation results addressing this point. If we agree that the critical question is: "How does CPT-characterized distortion provide an *additional meaning* to the suboptimal gap?  then a natural follow-up question arises:  **"Does CPT-characterized distortion offer any *benefit* to the agent in finding the optimal policy?"**
>
> To explore this, we conducted additional experiments. Consider an 8x8 Grid World where the agent always starts at $(0, 0)$ and can reach one of six different goal states located at $(4, 4)$, $(5, 5)$, $(6, 6)$, $(7, 7)$, $(8, 8)$, and $(9, 9)$. Each goal has a distinct reward from the set $[0.1, 1, 10, 100, 1000, 10000]$, and the agent incurs a step penalty of $-0.1$ for each move. Naturally, the optimal policy under real-world conditions would be to reach the goal at $(9, 9)$, which provides the highest reward.
>
> Interestingly, our simulation results reveal that there exists a **specific pair of reward distortion and probability distortion** (as defined in Definitions 1 and 2) that enables the agent to discover a better policy than what was optimized under real-world conditions.

---

> ### Author Response · Authors · 2024-11-26
>
> ### > Experimental Setup
>
> - **Distorted Worlds:** We consider six distorted environments, all of which share the same reward distortion pattern that adheres to Definitions 1 and 2.
> - **Distortion Parameter ($\gamma$):** $\gamma$ represents the distortion rate of transition probabilities, where smaller values of $\gamma$ indicate larger distortions.
> - **Algorithm:** All experiments were performed using the Q-learning algorithm.
> - **Outcome Metric:** The probabilities represent the visitation frequency of each goal state.
>
> ### > Results
>
> The results demonstrate that CPT-distorted worlds can enhance policy selection by leveraging the interaction between reward and probability distortions, ultimately leading to improved agent behavior in some cases.
> ### Table1
> | visitation probability of Goal State      | (4,4)   | (5,5)   | (6,6)   | (7,7)   | (8,8)   | (9,9) = Optimal goal   |
> |-----------------|---------|---------|---------|---------|---------|---------|
> | **(1) Real world (no distortion)**                   | 0.87929 | 0.09817 | 0.01621 | 0.00414 | 0.00168 | **0.00051** |
> | **(2) Distorted world (only reward distortion)** | 0.93762 | 0.05177 | 0.00728 | 0.00207 | 0.00090 | **0.00036** |
> | **(3) Distorted world (gamma=0.9 with reward distortion)**            | 0.92069 | 0.06538 | 0.00986 | 0.00264 | 0.00108 | **0.00035** |
> | **(4) Distorted world (gamma=0.8  with reward distortion)**            | 0.90552 | 0.07670 | 0.01244 | 0.00342 | 0.00147 | **0.00044** |
> | **(5) Distorted world (gamma=0.7  with reward distortion)**            | 0.87729 | 0.09861 | 0.01674 | 0.00482 | 0.00185 | **0.00070** |
> | **(6) Distorted world (gamma=0.6  with reward distortion)**            | 0.83943 | 0.12043 | 0.02770 | 0.00843 | 0.00309 | **0.00090** |
> | **(7) Distorted world (gamma=0.5  with reward distortion)**            | 0.81693 | 0.13363 | 0.03255 | 0.01048 | 0.00466 | **0.00175** |
>
> #### **Key Insights from Table 1:**
> - Results in **Table 1** show that as the level of distortion increases (moving from (2) → (3) → ... → (7)), the probability of visiting the optimal state $(9,9)$ also increases, surpassing the real-world case (denoted as (1)).
> - **Why does this happen?** : Intuitively, this occurs because **reward distortion alone** causes the agent to perceive the step reward of $-0.1$ as a significantly lower value, such as $-5$. This perception incentivizes the agent to prioritize shorter trajectories to avoid accumulating large penalties from step rewards, potentially leading to suboptimal goals (that is going to among (4,4) to (8,8).) However, **distorting probabilities** introduces exploration, encouraging the agent to consider a broader range of trajectories. This additional exploration allows the agent to better evaluate distant goals, often leading to improved policies.  Therefore, our answer to the above question is
>
> **[Answer]: CPT-distortion provide an intrinsic motivation (or help shape a beneficial intrinsic reward) for the agent to find the optimal policy that that of real-world**
>
> We have checked that CPT distribution can effectively serve this purpose. Specifically, we find that there exists a **reward bonus term** based on count-based exploration, expressed as:
>
> $$r(s,a) + \frac{\beta}{\sqrt{n(s, a)}}, $$
>
> where $n(s, a)$ represents the visitation count of state-action pair $(s, a)$.
>
> ### Table 2
> | visitation probability of Goal State          | (4,4)   | (5,5)   | (6,6)   | (7,7)   | (8,8)   | (9,9)   |
> |-------------------|---------|---------|---------|---------|---------|---------|
> | **(1) Real world (no distortion)**                   | 0.87929 | 0.09817 | 0.01621 | 0.00414 | 0.00168 | 0.00051 |
> | **(8) Real world (Reward Bonus - beta=0.5)**                   | 0.86549 | 0.10520 | 0.02041 | 0.00587 | 0.00235 | 0.00068 |
> | **(9) Real world (Reward Bonus  - beta=1.0)**                   | 0.85266 | 0.11078 | 0.02397 | 0.00817 | 0.00340 | 0.00103 |
> | **(10) Real world (Reward Bonus  - beta=1.5)**                   | 0.83812 | 0.11758 | 0.02782 | **0.01000** | **0.00479** | **0.00170** |
> | **(11) Real world (Reward Bonus  - beta=2.0)**                   | 0.82782 | 0.12016 | 0.03137 | 0.01253 | 0.00587 | 0.00215 |
> | **(12) Real world (Reward Bonus  - beta=3.0)**                   | 0.80600 | 0.12934 | 0.03732 | 0.01605 | 0.00831 | 0.00298 |
> | **(7) Distorted world (gamma=0.5)**            | 0.81693 | 0.13363 | 0.03255 | **0.01048** | **0.00466** | **0.00175** |
>
> #### **Key Insights from Table 2:**
> - Results in **Table 2** empirically show that increasing the portion of an exploration bonus (a common method for designing intrinsic rewards/motivation [4]) also increases the visitation probability of $(9,9)$ (moving from (8) → (9) → ... → (12)).
> - Notably, **case (10) matches with case (7)**, providing strong evidence that CPT-distortion can effectively serve as an intrinsic motivation mechanism for the agent to find the optimal policy.

---

> ### Author Response · Authors · 2024-11-26
>
> ### **[Q2] NASA example.**
> This example is for the "Q1. Real-world examples that can benefit from the definition of S-Black Swan events". That is more focused on supporting our hypothesis: the existence of a static black swan due to human misperception. Rather than NASA example, we address how the Healthcare example can be framed in an ML setting in the following section.
>
> ### **[Q3] Healthcare example.**
> Thanks for asking how this example relates to the ML setting. This comment is also similar to our comments of "[Weakness 2]. Distortion Can Also Be Regarded as Non-Stationarity" to Reviewer 234c.   We elaborate further as follows.
>
> Suppose we aim to predict a patient’s blood sugar level, denoted as $A$, as a function of factors $W$, $X$, $Y$, and $Z$—such as the level of carbohydrate intake or the severity of the patient’s illness. We design a prediction function:
>
> $$ A = f(W, X, Y, Z), $$
>
> and train $f$ on a given dataset of input-output pairs, $((w, x, y, z), a)$.
>
> However, due to human misperception about the relevant factors, we might overlook an important variable, $V$, which significantly impacts blood sugar spikes (i.e., black swan events). Specifically, humans might misperceive the joint probability $p(V, A)$ as being zero, and incorrectly assign the reward $r(V, A)$ as $r(V, A) + 100$, perceiving it as a benign event. As a result, $V$ is excluded when designing the function $f$.  Despite our best efforts, failing to include $V$ in the model design introduces errors or non-stationarity in the predicted blood sugar levels, particularly for extreme cases.
>
> This example illustrates how missing critical factors due to human misperception can lead to inaccurate predictions. Furthermore, addressing such misperceptions could potentially reduce the variance (or non-stationarity) of the output $A$, improving model reliability and performance.

---

> ### Author Response · Authors · 2024-12-02
> **Thanks for the feedbacks!**
>
> Dear Reviewer WW6X,
>
> Thank you for your thoughtful feedback, especially on how the results of this paper can contribute to the AI community  -- seems to be a critical consideration for this foundational draft. We hope our follow-up comments have effectively addressed your concerns. As the discussion phase draws to a close, we wanted to check if you have any additional feedback or concerns. Please feel free to share, and we would be more than happy to discuss further. Thank you once again!
>
> Best,
> Authors

---

### Official Review · Reviewer_DuS7 · 2024-11-08

**Soundness:** 3
**Presentation:** 2
**Contribution:** 2
**Rating:** 6
**Confidence:** 2

**Summary:**

This paper introduces the concept of "s-Black Swan" - statistically rare, high-risk events that can occur in unchanging environments due to human misperception of event probabilities and values. Authors present a formal framework from to define this in the context of MDPs and argue that for safety in RL systems, it's important to consider stationary MDPs that present these Black Swan events.

**Strengths:**

- This paper presents a new view point on how to look at Black Swan events. Specifically, it points to the case of stationary MDPs where agents have distorted perspective on reward signals and visitation probabilities which are likely to be overlooked by researchers.

- The mathematical rigor is strong - the authors have done a great job at defining s-Black Swan using the existing concepts of MDP and its special cases. It gives a good framework for future researchers to build upon while trying to model such s-Black Swan events. Specifically, theorem 5 seems to be the most useful aspect of this paper which provides an analytical bound of encountering the rare event with event probability.

Particularly, I liked the formulation of HEMDPs - seems to be a particularly useful modeling strategy.

**Weaknesses:**

- While the paper is very rigorous, the details might be very hard to follow for non-specialists. Some of the aspects are not very intuitive.

- This paper lacks a practical application demonstration - it'll be great if the authors can describe how a practitioner can use the definitions that the paper provides for a practical applications.

- Building upon the previous point - it'll be useful for us to understand how frequent are such MDPs where the users have a distorted view of the reward signals and the visitation probabilities.

**Questions:**

- Can you explain why the focus is on visitation probabilities v/s transition function? This is something that's not very intuitive.

- Why is the value distortion function and the probability distortion function modeled in such a piece-wise way? Is this just for simplicity? How do we decide how to model them?

- Example 2, case 3 - what does it mean for an MDP to be always a black swan?

- What is the significance of Lemma 1? Can you describe how should one intuitively understand it?

- Can you describe some applications where this will be useful? And how should one think about modeling such scenario - my understanding is that HEMDPs would be most appropriate for such situations.

---

> ### Author Response · Authors · 2024-11-19
> **Thanks for the comments!**
>
> Dear Reviewer DuS7,
>
> First, we would like to thank the reviewer for fruitful comments. Before addressing the reviewer's concerns, we kindly encourage the reviewer to first read the ***global comment on the main message***, followed by our responses that address specific weaknesses and questions.
>
> We are happy to provide further clarification on any remaining concerns or questions from the reviewer!
>
> **Q1. Why does distortion occur on visitation probability rather than transition probability?**
>
> Thank you for raising this excellent question! We carefully scrutinized which probability is appropriate to distort, and here is the reasoning behind our choice of visitation probability.  ***The primary reason lies in the definition of the event unit as a state-action pair $(s, a)$***. In prospect theory, the distortion of rewards (how good or bad an outcome is perceived) and probabilities occurs at the level of events $e$, specifically on $R(e)$ and $P(e)$. Translating this perspective to the context of a Markov Decision Process (MDP), the goodness or badness of a policy is revealed when a state $s$ is encountered, and the subsequent action $a$ is taken, yielding a reward $r(s, a)$. Therefore, it is natural to define the event unit as $(s, a)$ in the MDP framework.
>
> Consequently, probability distortion should occur where probabilities are defined over the support of $\mathcal{S} \times \mathcal{A}$. If we were to define distortion on transition probabilities, this would imply distorting the probabilities of the *next state* given a fixed $(s, a)$, which is defined over the support of $\mathcal{S}$. Such an approach would not align with the event-level distortion described in prospect theory.
>
> ***By distorting visitation probabilities, we ensure consistency with the literature on prospect theory and maintain alignment with the event-based perspective.***
>
>
> **Q2. Why are value distortion and probability distortion piecewise?**
>
> Thank you for raising this important issue. ***The piecewise nature of these distortions aligns with prospect theory, which models the irrationality of human behavior.*** While we have detailed how Definitions 1 and 2 were derived in lines 167 to 171, we will briefly summarize for clarity.
>
> The value function $v$ is piecewise due to *loss aversion*. For example, losing 1M feels significantly more impactful than gaining 1M, despite their equal absolute values. This asymmetry reflects the principle that losses are perceived more strongly than equivalent gains. Similarly, the probability distortion $p$ is piecewise because humans *overestimate or underestimate low-probability events* depending on whether they involve gains or losses. For instance, people overestimate the chances of winning the lottery (a low probability gain) but underestimate the likelihood of rare negative events, such as airplane crashes or pandemics like COVID-19.
>
> For further discussion, please see [Appendix C.2. Irrationality due to subjective probability].
>
>
> **Q3: Example 2 - Case 3**
>
> In the context of a stationary MDP, any $(s, a, t_{bs})$ that is identified as a black swan will always qualify as an S-black swan.
>
>
> **Q4: Importance of Lemma 1**
>
> Thank you for highlighting this point. **The importance of Lemma 1 lies in connecting distortion in visitation probability with real-world data collection, specifically in demonstrating how the draft's analysis can be applied in an experimental setting.**  Consider a scenario where an agent collects a dataset $\mathcal{D} = \{(s_i, a_i, r_i, s'_i)\}, i \in [N]$, where both the rewards $r_i$ and the state-action occupancy measure of $\mathcal{D}$ are subject to distortion. Simply distorting visitation probability does not intuitively explain how the data is collected. The data collection process follows a sequence: given a state and action, the agent receives a reward, transitions to the next state, and then selects the next action, repeating this process $N$ times. Lemma 1 addresses this by demonstrating that it is always possible to identify a distorted state that produces an equivalent distortion to that caused by the distortion function $w$. This result provides a more intuitive understanding of how the collected data exhibit distortion in an experimental setting, making the analysis more practical and relatable.
>
> **Q5, W2: Some Applications**
>
> Thank you for highlighting the potential future extensions of how this analysis can be utilized. We have addressed this in our response to **[Q4: Aiding Algorithm Design] from reviewer 234c**. Briefly, our work suggests that the analysis can inform the development of safe exploration strategies that account for human misperception and adapt to changes in probability or feasible set size, helping to better prevent S-black swan events.
>
> **W3: Frequency of black swans**
>
> Thank you for your question. Due to space limitations, we kindly refer you to our response to **[Q5: Hitting Time Analysis] from reviewer 234c** for further details.

---

> ### Author Response · Authors · 2024-11-23
> **Further Clarifications on the Reviewer's Concerns**
>
> Dear Reviewer DuS7,
>
> We sincerely hope our response has effectively addressed the reviewer’s initial concerns! Should Reviewer DuS7 have any further questions or points for discussion, we warmly encourage you to share them with us. Your thoughtful feedback has been instrumental in helping us:
>
>  * clarify the definitions and motivations of the value & visitation probability distortion functions, as well as emphasize the significance of Lemma 1,
>  * present empirical evidence (a gridworld multi-goal experiment) that strengthens our main message, and
>  * provide an additional real-world example illustrating how a static misperception can give rise to black swans and inform future algorithm design.
>
> We look forward to hearing whether these revisions have sufficiently addressed your concerns. We would be delighted to continue the discussion and provide further clarification to ensure a clear understanding of our work!
>
> Best, Authors

---

> > ### Comment · Reviewer_DuS7 · 2024-11-26
> >
> > Thanks for addressing my comments. I'd highly recommend the authors to include this discussion in their next version of the manuscript given that it contains useful clarifications. I'd be raising my score based on your comments.

---

> ### Author Response · Authors · 2024-12-02
> **Thanks for comments!**
>
> Dear Reviewer 2mu5,
>
> Thank you for your feedback on emphasizing the significance of the theorems and including examples of real-world cases of S-Blackswan to illustrate their practical applications. We wanted to follow up and inquire if you have any further concerns or feedback that we can address as we approach the end of the rebuttal phase. We greatly value your insights and are always happy to discuss further!
>
> Best, Authors

---

### Official Review · Reviewer_2mu5 · 2024-11-09

**Soundness:** 3
**Presentation:** 4
**Contribution:** 3
**Rating:** 6
**Confidence:** 3

**Summary:**

This paper challenges the conventional understanding of black swan events, which are typically seen as arising from unpredictable and dynamic environments. The authors propose that such high-risk, statistically rare events can also occur in static environments due to human misperception of events’ values and likelihoods, introducing the concept of S-BLACK SWAN. The paper categorizes black swan events, formalizes their definitions mathematically, and provides a framework for understanding and preventing these events through improved perception.

**Strengths:**

- The paper presents a novel hypothesis that black swan events can occur in static environments due to human misperception, which is a significant departure from the traditional view. This new perspective could open up fresh avenues for research in risk management and machine learning.
- The theoretical framework is well-developed, with rigorous mathematical formalizations and proofs. The use of Markov Decision Processes (MDPs) to model human perception and misperception is particularly robust.
- The paper is well-structured, with clear definitions and logical progression of ideas.
- By redefining the origins of black swan events, the paper has the potential to significantly impact the fields of machine learning, risk management, and decision theory. It provides a foundation for developing algorithms that can better handle rare, high-risk events.

**Weaknesses:**

- The paper lacks empirical validation of the proposed hypothesis. While the theoretical framework is strong, it would benefit from experimental results or real-world case studies demonstrating the occurrence of S-BLACK SWAN events.
- The mathematical formalizations, while rigorous, are quite complex and may be difficult for practitioners to apply directly. Simplifying some of the models or providing more intuitive explanations could enhance accessibility.
- The paper primarily focuses on financial and autonomous systems. Expanding the discussion to other domains where black swan events are critical, such as healthcare or environmental science, could broaden the impact of the work.

**Questions:**

- Can the authors provide empirical evidence or case studies that demonstrate the occurrence of S-BLACK SWAN events in real-world scenarios and model it using the proposed algorithm?
- How can the proposed algorithm be applied to other domains beyond finance and autonomous systems? Are there specific examples or case studies in areas like healthcare or environmental science?
- What are the next steps for developing algorithms based on the proposed hypothesis? Are there any preliminary results or ongoing projects that the authors can share?

---

> ### Author Response · Authors · 2024-11-19
> **Thanks for the comments!**
>
> Dear Reviewer 2mu5,
>
> We would like to sincerely thank the reviewer for inquiring about empirical evidence and the potential next steps of this work. Before addressing the specific concerns, we kindly encourage the reviewer to first review the ***global comment on the main message***, followed by our detailed responses to specific weaknesses and questions.
>
> We are happy to provide further clarification on any remaining concerns or questions from the reviewer!
>
> **Q1 and W1. Empirical evidence or case studies**
>
> Thank you for highlighting the importance of evidence. In addition to the Lehman Brothers bankruptcy case mentioned in the introduction, we present another case that illustrates how high-risk rare events (black swans) can occur, not due to unpredictable changes in the environment, but rather due to misperception - specifically, underestimating the probability of a certain event:
>
> * **Unexpected Drowning of NASA Astronauts Due to Overlooked Details**:
>   (The following is a summary of the case; please refer to [1, 2] for a more detailed account.) Before launching rockets, NASA conducted tests on its space suits using high-altitude hot-air balloons. On May 4, 1961, Victor Prather and another pilot ascended to 113,720 feet to evaluate the suit's performance. While the test itself was successful, an unforeseen risk led to tragedy during the planned ocean landing. Prather opened his helmet faceplate to breathe fresh air, and as he slipped into the water while attaching to a rescue line, his now-exposed suit filled with water. Despite NASA’s rigorous planning and preparation, the risk of opening the faceplate - perceived as an extremely minor detail - was **underestimated, resulting in catastrophic consequences**. This highlights how rigorously meticulous planning can still fail to account for overlooked events.
>
> *References:*
> [1] Jan Herman, “Stratolab: The Navy’s High-Altitude Balloon Research,” lecture, Naval Medical Research Institute, Bethesda, MD, 1995, archive.org/details/StratolabTheNavysHighAltitudeBalloonResearch.
> [2] Douglas Brinkley, American Moonshot (New York: Harper, 2019), 237.
>
> **Q2. Application of proposed viewpoints to other domains such as healthcare or environmental science**
>
> Thank you for pointing this out. We provide examples of how black swan events can occur in the domains of healthcare  as follows:
>
> * **Healthcare**:
>   Diabetes patients typically experience a highly chronic condition, making their state relatively predictable a few hours into the future (stationary environment). However, **rare hypoglycemic events**, characterized by a sudden and dangerous drop in blood sugar, pose significant risks. To address this, [3] developed an RL model capable of predicting changes in a patient's condition and providing optimized treatments. Furthermore, [4, 5] emphasize the critical importance of selecting appropriate signals as inputs, as **human misperceptions about what constitutes important signals** can lead to unexpected and suboptimal decisions. As a result, traditional supervised learning methods, such as general Transformers paired with loss functions like MSE, may struggle to accurately predict rare events like hypoglycemic episodes when the input signals fail to align with the underlying dynamics.
>
> *Reference*:
>
> [3] Wang, G., Liu, X., Ying, Z. et al. Optimized glycemic control of type 2 diabetes with reinforcement learning: a proof-of-concept trial. Nat Med 29, 2633–2642 (2023). https://doi.org/10.1038/s41591-023-02552-9.
> [4] Panda, D., Ray, R., & Dash, S.R. (2020). Feature Selection: Role in Designing Smart Healthcare Models. Intelligent Systems Reference Library.
> [5] C. Ambhika, S. Gayathri, A. T. P, B. G. Sheena, N. M and S. S. R, "Enhancing Predictive Modeling in High Dimensional Data Using Hybrid Feature Selection," 2024 5th International Conference on Electronics and Sustainable Communication Systems (ICESC),
>
> **Q3 and W3. Next steps on how to develop the algorithm**
>
> Thank you for inquiring about potential extensions of this work. The theorem we have developed can significantly inform future algorithm design. We would like to note that our response to Q3 same with our answer to **[Q4: Aiding Algorithm Design] from reviewer 234c**.
>
> **W2. difficult for practitioners to apply directly due to complex mathematical formalizations.**
>
> Thank you for raising this important concern. We recognize that, while rigorous, the mathematical formalizations may be challenging for practitioners to apply directly. In revision, we will provide intuitive explanations, such as real-world examples (such as NASA case and healthcare case mentioned above), analogies, or visual aids, which could make the concepts more accessible.

---

> ### Author Response · Authors · 2024-11-23
> **Further Clarifications on the Reviewer's Concerns**
>
> Dear Reviewer 2mu5,
>
> We hope our response helps address the reviewer’s initial concerns. If Reviewer 2mu5 has any additional points or questions to discuss, please do not hesitate to let us know. Your insightful comments have guided us toward
>  * presenting empirical evidence (a gridworld multi-goal experiment) that supports our main message,
>  * providing an additional real-world case that interprets the origin of the black swan as a static misperception
>
> and we look forward to hearing whether this has resolved your concerns. We would be delighted to continue the discussion and provide further clarification to ensure a clear understanding of our work.
>
> Best,
> Authors

---

> > ### Comment · Reviewer_2mu5 · 2024-11-29
> > **Response to authors**
> >
> > Thank you for the response, which addresses my concerns. I appreciate the author's effort in modeling the S-black-swan events. Thus, I increase the score to 6. However, I believe that this work would be more beneficial if it proposed algorithm(s) or method(s) to mitigate the potential risks of these events.

---

> ### Author Response · Authors · 2024-12-02
> **Thanks for the feedback!**
>
> Dear Reviewer 2mu5,
>
> Thanks for your feedback and for suggesting potential futures, such as algorithms to mitigate the potential risks of these events (correcting the human misperception). We wanted to follow up and inquire if you have any further concerns or feedback that we can address as we approach the end of the rebuttal phase. We greatly value your insights and are always happy to discuss further!
>
> Best,
> Authors

---

### Author Response · Authors · 2024-11-19
**Main Message**

We would like to sincerely thank all the reviewers for their thorough and insightful feedback! We hope that this response effectively delivers the main message of our draft and contributes to a clearer and more comprehensive understanding of our work.

**Our central argument highlights that if black swan events can also arise from human misperception, AI safety algorithms should broaden their focus beyond solely developing accurate predictive models for rare events.** Specifically, they should also aim to address and correct biases in data generated by human behavior, which are shaped by such misperceptions.  This perspective emphasizes the importance of prioritizing **not only algorithmic design**, which is the current standard approach in AI safety, but also the mitigation of human-induced biases in the **data**. We believe that introducing a new direction for AI safety algorithms - one that has not been widely explored - is vital for the AI community. A deeper understanding of the origins of black swans paves the way for achieving greater control and making more informed, optimal decisions.


We appreciate that this paper primarily focuses on definitions and offers a viewpoint that we believe is essential for guiding the direction of future AI safety algorithm design, rather than proposing specific algorithms to correct human misperception. ***Its central goal is to establish a foundational understanding of how black swans may be interpreted as a result of human misperception.*** While we acknowledge that the paper may not meet the standard for acceptance if it does not provide sufficient evidence to support this interpretation in the context of human perception, we also see significant potential in its contributions.

If the paper presents a well-reasoned perspective on the possibility of black swans occurring in stationary environments due to human misperception, supported by convincing evidence, we believe it could positively influence the AI community. Specifically, it could inspire efforts to develop AI safety algorithms that address human-biased data, moving beyond a sole focus on creating accurate forecasting algorithms for black swans. Exploring methods for bias correction or algorithm development would be a natural and meaningful extension of this work, although we recognize that such efforts fall outside the immediate scope of this paper.

---

### Author Response · Authors · 2024-11-21
**Experiments to Support Our Main Message**

Dear Reviewers 2mu5, DuS7, WW6X, and 234c,

We hope our comments address the reviewers' concerns! During the active discussions with Reviewer 234c, who raised *a very insightful question* to support the clarity and significance of this paper’s contribution, we focused on the following key question:

**"How does CPT-characterized distortion provide an additional meaning to the suboptimal gap?"**

To address this question, we conducted **simulation experiments in Gridworld.** We kindly recommend the reviewers refer to the section titled:  "[Weakness 1] S-black swan is just an emergence from a non-zero model error from true P and R, so the theoretical analysis is trivial, making it indistinguishable from prior RL theory literature focusing on PAC bound analysis via model estimation - (Additional Answer 2): Experiment to support our answer for [Weakness 1]"

This discussion is included in the **discussion panel with Reviewer 234c.**

Through the experiment, we concluded that:

**"CPT-distortion provides an intrinsic motivation (or helps shape a beneficial intrinsic reward) for the agent to discover the optimal policy, potentially outperforming that of the real-world scenario."**

These experimental results strongly support our main message: Interpreting the risk by human misperception reveals novel phenomena, underscoring the need for in-depth scrutiny in future algorithm design.

We hope our experiment results address the concerns of reviewers who were interested in seeing simulation results.  Thank you again for your thoughtful feedback and active engagement. We welcome any further questions or discussions!

Best regards,
Authors

---

### Meta-Review · Area_Chair_rVZy · 2024-12-24

**Metareview:**

Thank you for your submission to ICLR. This paper introduces the S-Black Swan (i.e., black swan events in stationary environments), which provides formalism for the setting where black swans occur in unchanging environments due to a misperception of events’ probabilities and values. This is in contrast to the typical assumption that a black swan event occurs in an unpredictable and dynamic environment. The authors define S-Black Swan in the context of a Markov Decision Process (MDP).

Reviewers agreed that the paper was clearly written, provides a novel contribution, and has strong mathematical formalism (which captures intuition well). However, there were concerns from multiple reviewers about whether this paper was well-enough motivated from problems in machine learning, and in general whether it has enough connection to the broader machine learning community—and wanted a clear explanation of how results from this paper can benefit this community. Also, multiple reviewers requested an empirical/practical application demonstration to make this connection more clear. In their rebuttal, the authors put a lot of effort into justifying this connection, and giving concrete demonstrations of applications, along with an experimental result. On balance, a majority of reviewers updated their scores by the end of the rebuttal period, and felt that their concerns were addressed. I therefore recommend this paper for acceptance, and encourage the authors to keep this feedback in mind when putting together the camera-ready version of their paper.

**Additional Comments On Reviewer Discussion:**

During the rebuttal period, the authors made efforts to thoroughly answer the reviewers’ questions, where they focused on explaining the practical application of their method and providing empirical results. There was a healthy discussion with multiple reviewers, which led to multiple reviewers raising their scores. In the end, the authors addressed most concerns from all of the reviewers.

---

### Decision · Program_Chairs · 2025-01-22

Accept (Poster)